# Multiregional single-cell dissection of tumor and immune cells reveals stable lock-and-key features in liver cancer

Lichun Ma [1,2,14], Sophia Heinrich[1,3,14], Limin Wang[1], Friederike L. Keggenhoff[4], Subreen Khatib[1], Marshonna Forgues[1], Michael Kelly [5], Stephen M. Hewitt [6], Areeba Saif[7], Jonathan M. Hernandez[7], Donna Mabry[8], Roman Kloeckner [4,9], Tim F. Greten [8,10], Jittiporn Chaisaingmongkol [11,12], Mathuros Ruchirawat [11,12], Jens U. Marquardt [4,13] & Xin Wei Wang [1,10] ✉

Intratumor heterogeneity may result from the evolution of tumor cells and their continuous interactions with the tumor microenvironment which collectively drives tumorigenesis. However, an appearance of cellular and molecular heterogeneity creates a challenge to define molecular features linked to tumor malignancy. Here we perform multiregional single-cell RNA sequencing (scRNA-seq) analysis of seven liver cancer patients (four hepatocellular carcinoma, HCC and three intrahepatic cholangiocarcinoma, iCCA). We identify cellular dynamics of malignant cells and their communication networks with tumor-associated immune cells, which are validated using additional scRNA-seq data of 25 HCC and 12 iCCA patients as a stable fingerprint embedded in a malignant ecosystem representing features of tumor aggressiveness. We further validate the top ligand-receptor interaction pairs (i.e., LGALS9-SLC1A5 and SPP1-PTGER4 between tumor cells and macrophages) associated with unique transcriptome in additional 542 HCC patients. Our study unveils stable molecular networks of malignant ecosystems, which may open a path for therapeutic exploration.

Multi-level analyses of human cancer tissues unveil a vast degree of molecular heterogeneity among cells within individual tumors, a feature known as intratumor heterogeneity (ITH)[1]. A varying degree of ITH can be found in most, if not all, major solid malignancies and these features are universally associated with patient's prognosis[2]. Tumor cell evolution may be a main contributor to ITH because each tumor cell or their-derived subclones compete with each other in an adverse milieu of the tumor microenvironment (TME) for survival, resulting in a complex tumor ecosystem, where tumor cells may serve as the architect to orchestrate various cell types in the tumor cell community

[1]Laboratory of Human Carcinogenesis, Center for Cancer Research, National Cancer Institute, Bethesda, MD 20892, USA. [2]Cancer Data Science Laboratory, Center for Cancer Research, National Cancer Institute, Bethesda, MD 20892, USA. [3]Clinic for Gastroenterology, Hepatology and Endocrinology, Hanover Medical School, Hanover, Germany. [4]Department of Medicine I, Lichtenberg Research Group, Johannes Gutenberg University, Mainz, Germany. [5]Frederick National Laboratory for Cancer Research, Leidos Biomedical Research, Inc., Frederick, MD 20701, USA. [6]Laboratory of Pathology, Center for Cancer Research, National Cancer Institute, Bethesda, MD 20892, USA. [7]Surgical Oncology Program, Center for Cancer Research, National Cancer Institute, Bethesda, MD 20892, USA. [8]Thoracic and GI Malignancies Branch, Center for Cancer Research, National Cancer Institute, Bethesda, MD 20892, USA. [9]Institute for Interventional Radiology, University of Lübeck, Lübeck, Germany. [10]Liver Cancer Program, Center for Cancer Research, National Cancer Institute, Bethesda, MD 20892, USA. [11]Laboratory of Chemical Carcinogenesis, Chulabhorn Research Institute, Bangkok 10210, Thailand. [12]Center of Excellence on Environmental Health and Toxicology, Office of Higher Education Commission, Ministry of Education, Bangkok 10400, Thailand. [13]Department of Medicine I, University Medical Center, Lübeck, Germany. [14]These authors contributed equally: Lichun Ma, Sophia Heinrich. ✉e-mail: xw3u@nih.gov

to facilitate its growth[3,4]. Because stromal and immune cells in the tumor ecosystem may compete with tumor cells for space and nutrients in addition to potentially activated immune surveillance to restrict tumor growth, tumor cells may develop unique features to reprogram the TME to evade from the competition[5]. Therefore, like the landscape in nature, the survival fitness of each tumor may be dictated by a unique tumor cell landscape shaped by the intrinsic tumor biology, tumor evolution and the TME, which gives rise to an appearance of a complex tumor ecosystem. Obviously, ITH represents a major barrier for implementing effective therapeutic interventions such as systemic therapies because of the difficulty in identifying stable molecular features linked to malignancy due to evolution-driven moving targets[4].

Overwhelming evidence indicates the remarkable heterogeneity of solid malignancies at the phenotypic and genetic levels. For example, somatic mutation analysis revealed that most hepatocellular carcinoma (HCC) cases examined show ITH at the genetic levels[6]. However, recent studies of lung cancer and leukemia with the application of single-cell technologies revealed that malignant clonal dominance is a cell-intrinsic and heritable property and transcriptome-based cellular state arises largely independently of genetic variation[7,8]. This raises questions about the accuracy in defining tumor cell biodiversity at the genetic levels due to the presence of passenger mutations and that transcriptomic heterogeneity may represent a good alternative to model cancer evolution[1,9]. The key question remains as how best to define biologically meaningful tumor ecosystem. Since tumor evolution is accompanied by continuous interactions of tumor cells and the TME, defining cellular features and its underlying molecular communication networks that shape tumor biology and consequently drive tumor evolution may be a key to improve molecular understanding of tumor ecosystems and to develop effective therapeutic approaches for solid tumors. Herein, we postulated that each tumor ecosystem may represent the success in enriching a unique combination of tumor-stromal interaction networks that promote tumor evolution under selective pressure. Defining the interactions of tumor and immune/stromal cells may, thus, represent unique fingerprints stable for its tumor biology, a feature analogous to the lock-and-key model that describes the enzyme-substrate interaction proposed by Emil Fischer over 127 years ago[10].

Cells are the smallest structural and functional unit of a tumor lesion. Therefore, tumor biology should be studied at the single-cell level[11,12]. Single-cell-based transcriptomic analysis has been increasingly used to study tumor and immune cell compositions in normal and diseased tissues such as liver cancer to better capture the tumor ecosystem[13,14]. As the incidence and mortality of HCC and intrahepatic cholangiocarcinoma (iCCA), the two main histological types of liver cancer, are still on the rise at the global level[15], several recent studies have exploited HCC and iCCA ecosystems by single-cell transcriptomic analysis[16–20]. These studies mainly relied on tumor biopsies obtained from a single region within the tumor to generate tumor cell composition, which have revealed vast ITH. It is unclear whether sampling bias influences the observed tumor ecosystem and the associated ITH within each tumor and consequently data interpretation, or if the tumor composition is relatively stable with a secured communication network of tumor and the TME that supports tumor malignancy.

In this study, we aim to determine the spatial distribution of tumor cells and TME by performing a multiregional single-cell RNA sequencing (scRNA-seq) analysis of HCC and iCCA from seven liver cancer patients and validate the results in single-cell data from an additional 37 HCC and iCCA patients. We find that while ITH is evident, variations of the tumor cell composition and the corresponding communication networks of tumor cells and the TME are smaller within each tumor than between tumors regardless of tumor size and corresponding distance among sampling tissues. We identify a fingerprint consisting of LGALS9-SLC1A5, SPP1-PTGER4 as tumor and macrophage-derived ligand–receptor interaction pairs, linked to

tumor aggressiveness. We independently validate the stability of the expression patterns and prognostic value of the LGALS9-SLC1A5 and SPP1-PTGER4 pairs using both bulk transcriptome profiles of 542 HCC samples from three independent cohorts and multiplexed fluorescence in situ hybridization-based profiles of 258 HCC samples from two cohorts. Our multiregional single-cell dissection of tumor and immune cells reveals a stable tumor-macrophage interaction network linked to ITH and HCC prognosis, which may provide the basis for further functional exploration including the development of rationale therapeutics for liver cancer.

## Results
### Multiregional liver tumor cell transcriptome profiles
To determine the spatial distribution of tumor cells and the TMEs as well as its stability within each tumor, we performed multiregional single-cell transcriptomic profiling of liver tumor specimens with varying tumor sizes from four HCC patients and three iCCA patients who underwent surgical resection (Supplementary Table 1). Specifically, we prepared single cells from five separate regions for each tumor, i.e., three tumor cores (T1, T2, and T3), one tumor border (B) and an adjacent normal tissue (N), followed by droplet-based 5′ scRNA-seq of these samples (Fig. 1a). We removed one sample (T3 from case 3C) due to single-cell library failure and thus a total of 34 samples were included in this study. We identified malignant and non-malignant cells by using the same method applied successfully in our previous studies and further used adjacent normal liver tissues as a control (Supplementary Fig. 1)[17,20]. Samples with >10 malignant cells detected were used for the analysis of malignant cells. With this criterion, six patients in our cohort had detectable malignant cells (Supplementary Fig. 1a). We then determined the similarity of tumor cell composition among multiple regions for each tumor from the six patients. t-distributed stochastic neighbor embedding (t-SNE) analysis revealed that malignant cells formed patient-specific clusters regardless of tumor regions (Fig. 1b and c), suggesting a much smaller interregional heterogeneity than intertumoral heterogeneity. This was also evident from hierarchical clustering analysis of multiple regions, where tumor cells from the same patient tended clustering together in the hierarchical tree (Fig. 1d). This was consistent with our previous bulk transcriptome study of HCC[21]. In contrast to patient-specific patterns of malignant cells, epithelial cells from adjacent normal tissues of different cases were mixed and separated from malignant cells (Supplementary Fig. 2), indicating shared non-malignant epithelial cell states among patients, which further served as a control to define malignant cells.

While tumor cells from multiple regions of each individual case were clustered together (Fig. 1c), we observed noticeable differences in tumor histology among different regions of each tumor, regardless of a difference in tumor size among these tumors (Fig. 1e and Supplementary Fig. 3). To quantitively determine the heterogeneity of tumor cells, we calculated intraregional heterogeneity as the distribution of pair-wise correlation of malignant cells within a specific region while interregional heterogeneity as the correlation of malignant cells among multiple regions within a patient. We used intertumoral heterogeneity among different patients as a reference. Noticeably, while some regional differences within each tumor were noted, correlations among intraregional and interregional tumor biopsies within each patient were much greater than that of intertumor among different patients (Fig. 1f). We also evaluated the inferred chromosomal copy number variations (CNVs) and found considerable distinct patterns among patients while much smaller differences among multiple tumor regions within each case (Supplementary Fig. 4a). To determine the impact of tumor size on tumor heterogeneity, we calculated its correlation with tumor heterogeneity among different regions and found no relationship in these samples (Supplementary Fig. 4b–d). These results indicate that while tumor size and histology may vary among patients, there was a much smaller

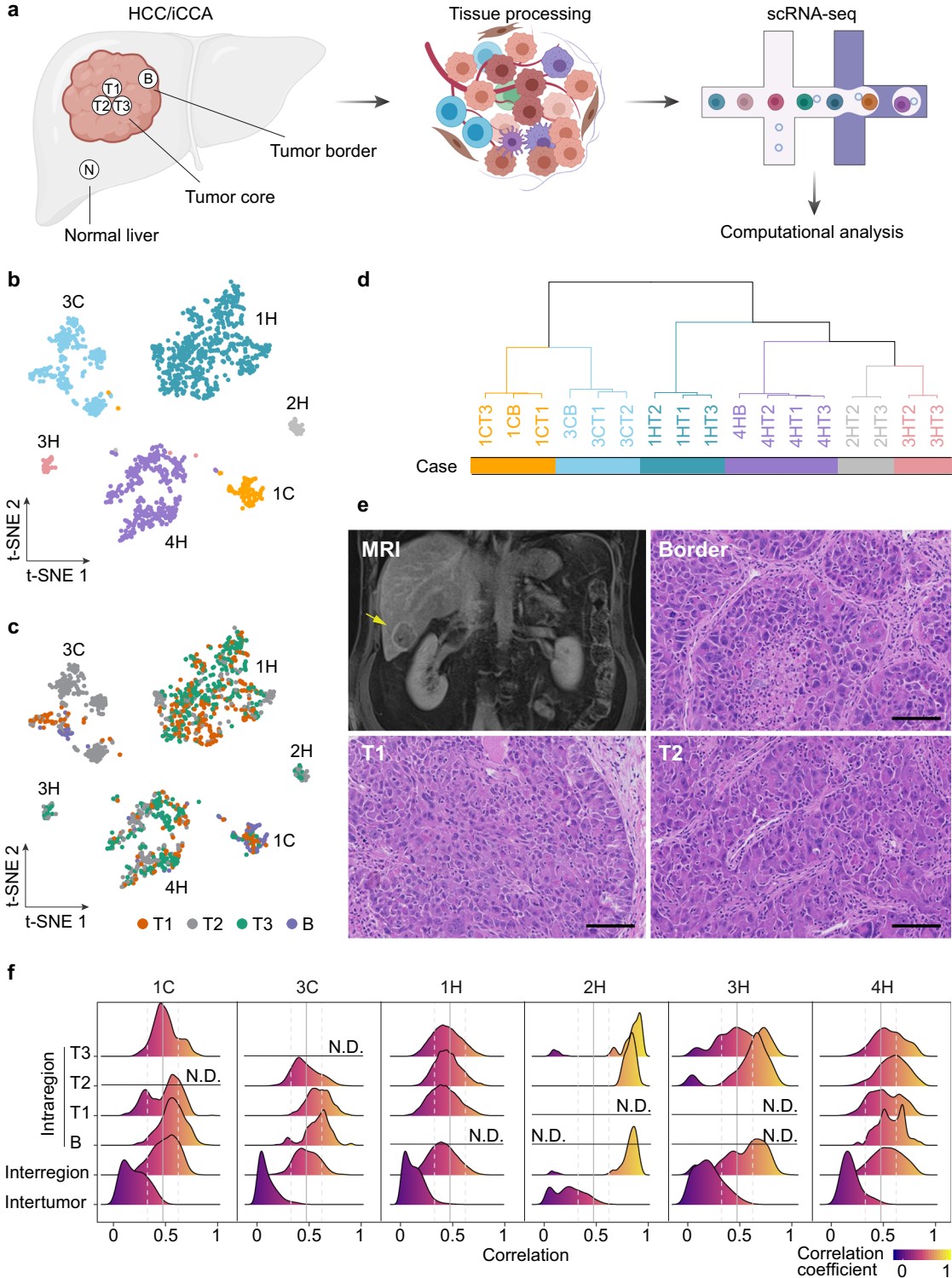

**Fig. 1 | Multiregional single-cell transcriptome profiling of liver cancer.**
**a** Workflow of multiregional tissue collection, processing, scRNA-seq, and data analysis. B, tumor border; T1, T2, and T3, three tumor cores; N, adjacent normal tissue. scRNA-seq, single-cell RNA sequencing. The figure was generated using BioRender. **b**, **c** t-SNE plot of malignant cells colored by cases (**b**) or tumor regions (**c**). Case ID was named according to the histological subtypes of HCC and iCCA. H, HCC; C, iCCA. B, tumor border; T1, T2, T3, three tumor cores. **d** Hierarchical clustering of malignant cells from each tumor region across all cases. Samples were named according to the histological subtypes and tumor regions. **e** Representative magnetic resonance imaging (MRI) of case 4H and histopathology of tumors from border, T1 and T2 of this case. Scale bars, 50 μm. Multiregional imaging pictures from all 7 cases are included as supplementary Figure 3. **f** The distribution of pairwise correlations of malignant cells within each tumor region (intraregion), across regions within each individual case (interregion) and across cases (intertumor). Pearson's correlation coefficient was applied. N.D., not detectable. Solid and dashed gray lines indicate the mean and standard deviation of all intraregional correlation values. Source data are provided as a Source data file.

difference in tumor cell transcriptomic activity within a patient than between patients. This observation is consistent with a recent report regarding HCC regional transcriptomic heterogeneity[22].

## Dynamics of transcriptomic heterogeneity of malignant and non-malignant cells

To evaluate the landscape of cellular dynamics of malignant cells from different regions linked to their transcriptomic heterogeneity, we performed trajectory analysis using RNA velocity[23,24], which determines cellular dynamics including developmental lineages and differentiation states based on splicing kinetics of tumor cell transcriptome. We found cellular trajectories of malignant cells again appear similar among different tumor regions within each tumor while heterogeneity in expressions of different stemness genes is noted. For example, the malignant cells from case 1C (7.5 cm in tumor size) followed a similar trajectory regardless of the tumor region sampled (Fig. 2a). When ranking tumor cells of 1C along their latent time estimated using RNA velocity with the expression of cell stemness related marker genes (i.e., EPCAM, KRT19, ICAM1)[16] and a tumor progression related gene (i.e., SPP1)[20], we found a similar gene expression pattern among sampling regions while varying expression among individual cells (Fig. 2b). Similar results were found in other cases (Supplementary Fig. 5). In addition, we performed fluorescence-activated cell sorting (FACS) analysis of EPCAM+ cells or GPC3+ cells for three HCC samples with available cryopreserved single-cell suspension, because the two markers were known elevated in tumor cells. The proportion of EPCAM+ or GPC3+ cells was relatively stable in the tumor core regions while differences were noted when compared to the tumor border regions, suggesting that the proportion of those cells may vary among tumor regions (Supplementary Fig. 6). We also confirmed that SPP1 expression is elevated in tumor cells but its expression is heterogeneous (Supplementary Fig. 7), which is consistent with previous publications[21].

We also determined the spatial landscape of non-malignant cells of HCC and iCCA by dimensional reduction using a manifold learning method of uniform manifold approximation and projection (UMAP)[25]. In contrast to the patient-specific patterns in malignant cells, non-malignant cells were mainly grouped based on their cell lineage, i.e., T cells, B cells, tumor-associated macrophages (TAMs), cancer-associated fibroblasts (CAFs), tumor-associated endothelial cells (TECs), hepatocytes and cholangiocytes, which were determined using lineage-specific marker genes (Fig. 3a, b). We observed similar cellular patterns of the TMEs among multiple tumor regions or among patients while a small difference was noted (Fig. 3c, d). When each cell type was analyzed separately, we found a high correlation of T cells from different sampling regions within each case, suggesting that the T-cell transcriptomic profiles appear stable among different sampling regions (Fig. 3e). We further evaluated the presence of CD3+ T cells using immunohistochemistry analysis, where relatively similar CD3+ T-cell numbers between tumor cores and tumor borders within each case than among cases was revealed with available paraffin blocks (Supplementary Fig. 8). In contrast, we found the correlation of CAFs, TECs, TAMs and B cells varied among different regions with some cases showing a high correlation while others showing a low correlation (Supplementary Fig. 9a). However, variations of immune/stromal cell types among different sampling regions were not correlated with tumor size, similar to the features of tumor cells (Supplementary Fig. 9b).

Based on the overall features of all cells in the TME, we observed immune activation in 3C and 3H while immune suppression in 4H and 1H (Supplementary Fig. 9c). For the rest of the cases, lack of the immune activities was observed (Supplementary Fig. 9c). Since T-cell profiles appeared most stable, we further determined T-cell subtypes and their composition in each tumor region. We identified 21 subsets and defined them based on the top differentially expressed genes

(Supplementary Fig. 10a and b). We observed similarity of the T-cell state composition in some tumor regions within each case while variations among others (Supplementary Fig. 10c). For example, 4H had the most stable T-cell subset composition among all the tumor regions, which is consistent with the highest correlation of T-cell transcriptomic profiles in this case (Fig. 3e and Supplementary Fig. 10c). In contrast, similarity of T-cell states was found in only two of the biopsied regions in the case of 1C. Thus, T-cell states appeared very dynamic although the difference within a patient seems smaller than between patients. Taken together, transcriptomic heterogeneity of malignant cells and immune/stromal cells appears smaller among sampling regions within each tumor lesion than among tumors from different patients while cellular heterogeneity within each tumor is noted. These results are consistent with recent publications in HCC, non-small-cell lung cancer and renal cell carcinoma using single-cell technologies[22,26,27].

## Multiregional tumor-immune communication networks

Given the noticeable histological and transcriptomic heterogeneity of a tumor lesion described above and elsewhere, it is imperative to identify stable molecular features linked to tumor biology. As tumor progression is a dynamic process involving continuous interactions of tumor cells and stromal/immune cells, tumor cells may profoundly influence various cell types within the tumor ecosystem to promote their own survival and the dissemination of malignancy[4]. Meanwhile, the TMEs including tumor-associated matrix suppress tumor growth by effective immune surveillance but may also be educated by tumor to support tumor progression[28–30]. We hypothesized that each tumor ecosystem may contain specific molecular communication networks between tumor cellular activities and the immune cell landscape unique to each patient, as these features may have minimum sampling bias. To identify the communication networks, we searched for ligand–receptor interactions of malignant cells and the TME within each patient using CellPhoneDB[31,32] (see "Methods"). We identified patient-specific interaction networks that are consistently well-conserved among different tumor regions from each individual patient (Fig. 4a, b, Supplementary Figs. 11–13), regardless of the direction of the interactions (i.e., tumor-to-TME, ligands from malignant cells and receptors from non-malignant cells; TME-to-tumor, ligands from non-malignant cells and receptors from malignant cells). These results indicate that the molecular communication networks appear relatively stable regardless of sampling locations, which likely reflects the intrinsic tumor biology of each tumor. Noticeably, the interactions are more stable in HCC than iCCA (Fig. 4c). We also determined the interactions between tumor cells and T-cell subsets, where relatively stable interactions were observed within each case (Supplementary Fig. 12b, c). In contrast to the consistency among different tumor regions within a specific patient, the ligand–receptor interaction pairs varied across patients, showing cell-to-cell communication networks unique to each patient (Fig. 4a and d). To determine whether patient-specific ligand–receptor interaction networks were mainly contributed by tumors or by TME, we developed a strategy by performing random shuffle of either tumor or TME (Fig. 4e). Not surprisingly, we found that the strength of the network interaction deteriorates faster for tumor random shuffle than TME random shuffle (Fig. 4f), suggesting the uniqueness is mainly controlled by malignant cells. Taken together, these results indicate that each tumor may contain relatively stable but unique communication networks between tumor and the TME, where malignant cells mainly contribute to the patient-specific patterns.

## The tumor-immune communication networks are associated with patient outcomes

To determine if patient-specific tumor-immune interaction networks are biologically important and are associated with tumor aggressiveness, we extended our search for ligand–receptor interactions in

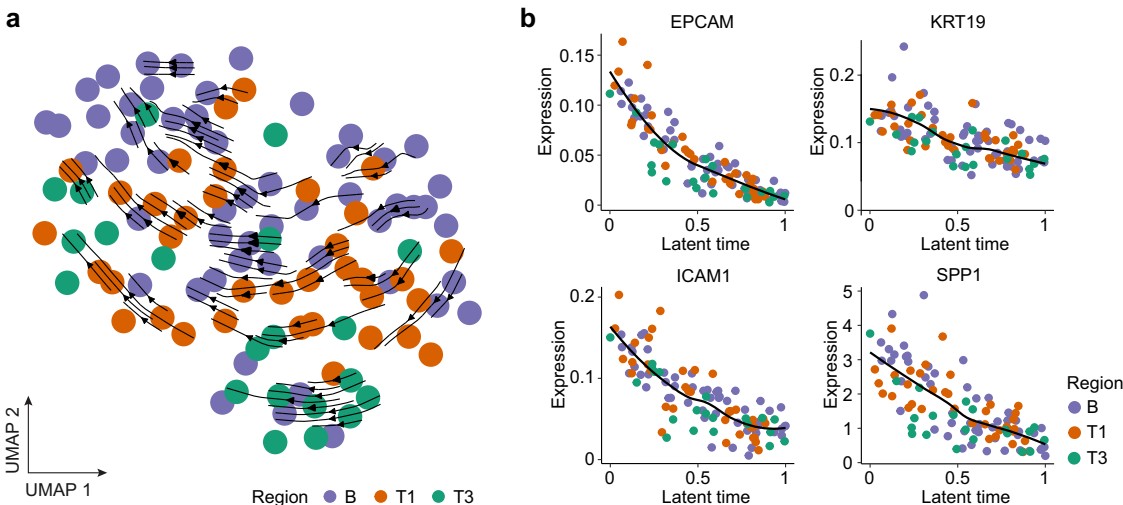

Fig. 2 | **Multiregional tumor cell trajectory of case 1C. a** RNA velocity of malignant cells from all tumor regions with viable malignant cells. **b** Expression of EPCAM, KRT19, ICAM1, and SPP1 in malignant cells along cellular latent time determined by RNA velocity method in (**a**).

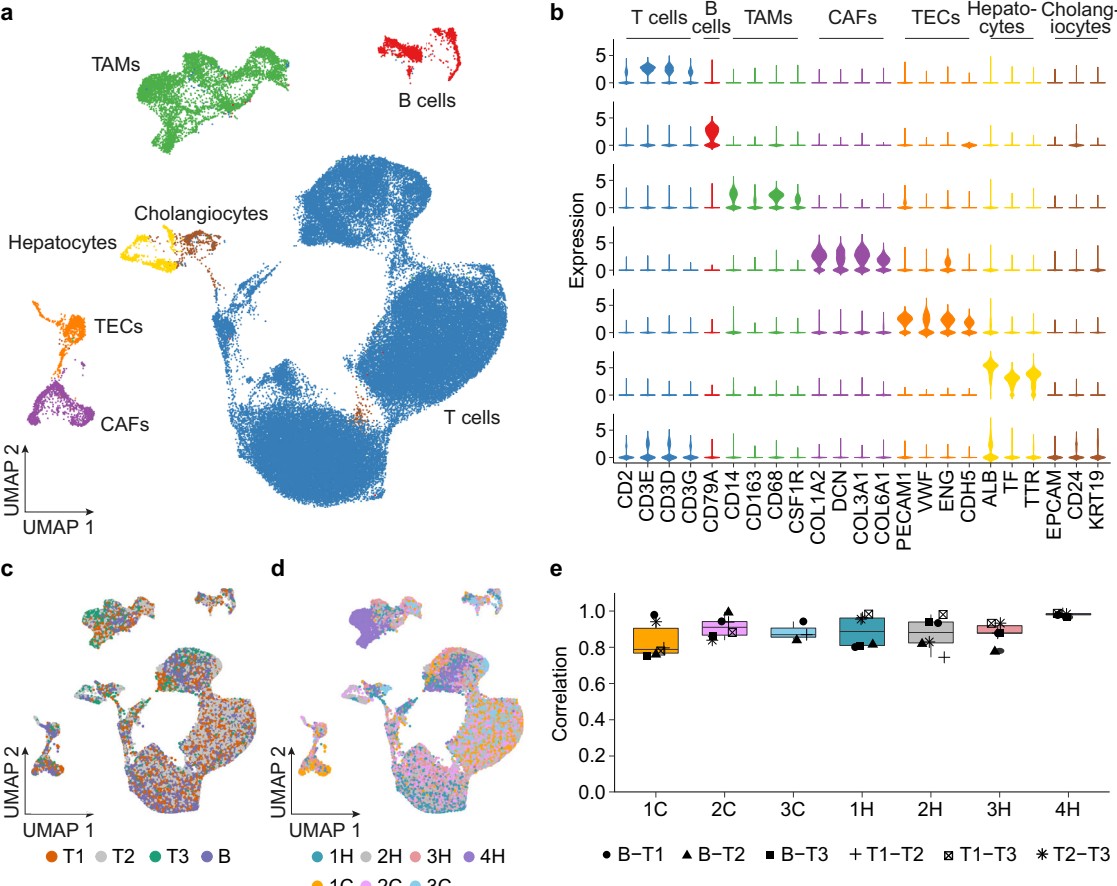

Fig. 3 | **Landscape of multiregional non-malignant cells. a** UMAP of non-malignant cells colored by cell types. CAFs, cancer-associated fibroblasts; TAMs. tumor-associated macrophages; TECs, tumor-associated endothelial cells. **b** Violin plots of cell-type specific marker gene expression in non-malignant cells. **c, d** UMAP of non-malignant cells colored by tumor regions (**c**) and case IDs (**d**). **e** Correlation of T cells ($n = 61,561$) between different tumor regions of each individual case. In the box plots, the central rectangles span the first quartile to the third quartile, with the segments inside the rectangle corresponding to the median. Whiskers extend 1.5 times the interquartile range. Source data are provided as a Source data file.

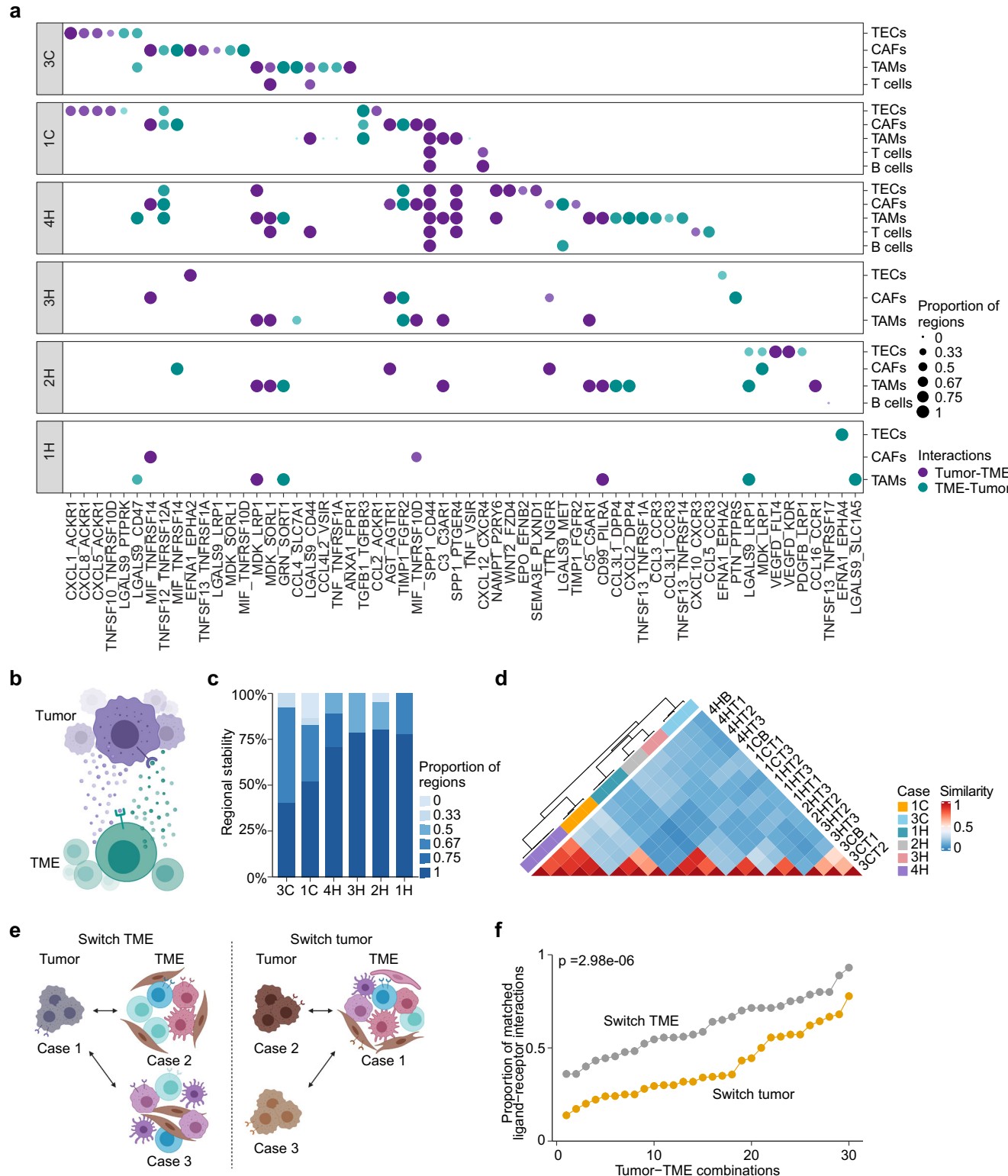

additional 46 tumor samples from 25 HCC and 12 iCCA patients described recently[20]. We included samples with > 15 detectable malignant cells per tumor for determining interactions of malignant cells and non-malignant cells. We further filtered the identified ligand–receptor pairs by using those found in the multiregional analysis (Fig. 4a, see "Methods"), which were selected using stringent criteria and demonstrated stable communication programs across multiple regions within each patient. We found two main clusters with different interaction networks based on a hierarchical relationship of the

ligand–receptor interaction activities (Fig. 5a). Noticeably, patients from the two clusters have significantly different overall survival, suggesting that the tumor-immune interaction networks are biologically distinct between the two clusters linked to tumor biology (Fig. 5b). Because this cohort contains both HCC and iCCA patients, we also analyzed HCC patients separately to avoid potential tumor type bias. We found a consistent trend of survival difference between the two clusters (Fig. 5c). We did not analyze iCCA patients separately as they were all in Cluster 1. To determine the key interactions of each cluster,

**Fig. 4 | Communication of malignant cells and non-malignant cells.**
**a** Ligand–receptor interactions of malignant cells and non-malignant cells in six cases with viable malignant cells. Each column indicates a ligand–receptor pair, with the first and the second gene representing a ligand and a receptor, respectively. Each row represents a non-malignant cell type that interacts with malignant cells. The direction of an interaction is indicated by colored dot. Purple, malignant cells provide ligands and interact with receptors from non-malignant cells in the TME; green, non-malignant cells in the TME provide ligands and interact with receptors from malignant cells. The size of each dot represents the proportion of tumor regions within each case in identifying a specific interaction pair, with 1 indicating occurrence in all tumor regions and 0 indicating occurrence in none of the tumor regions. **b** Illustration of ligand–receptor interactions of tumor and the TME. Purple dots, ligands from tumor; green dots, ligands from TME. **c** Stacked bar plot of the percentage of ligand–receptor pairs in each individual case found in

certain proportion of tumor regions. One means that a pair was found in all tumor regions within a case while zero means that a pair was found in none of the tumor regions. **d** Similarity of ligand–receptor interactions among multiple regions of different cases. Zero indicates no overlap of ligand–receptor interactions while 1 means a full overlap of ligand–receptor interactions between samples. **e** Illustration of switching TME and switching tumor. Switching TME indicates that tumor cells from one case are combined with TMEs from other cases to form distinct tumor ecosystems. Switching tumor indicates that TME from one case are combined with tumors from other cases to form distinct tumor ecosystems. **f** The proportion of matched ligand–receptor interactions from switching tumor or TME with the original search of using paired tumor and TME from the same case. Student's t-test (two-sided) was applied with *p* value provided. **b** and **e** were generated using BioRender.

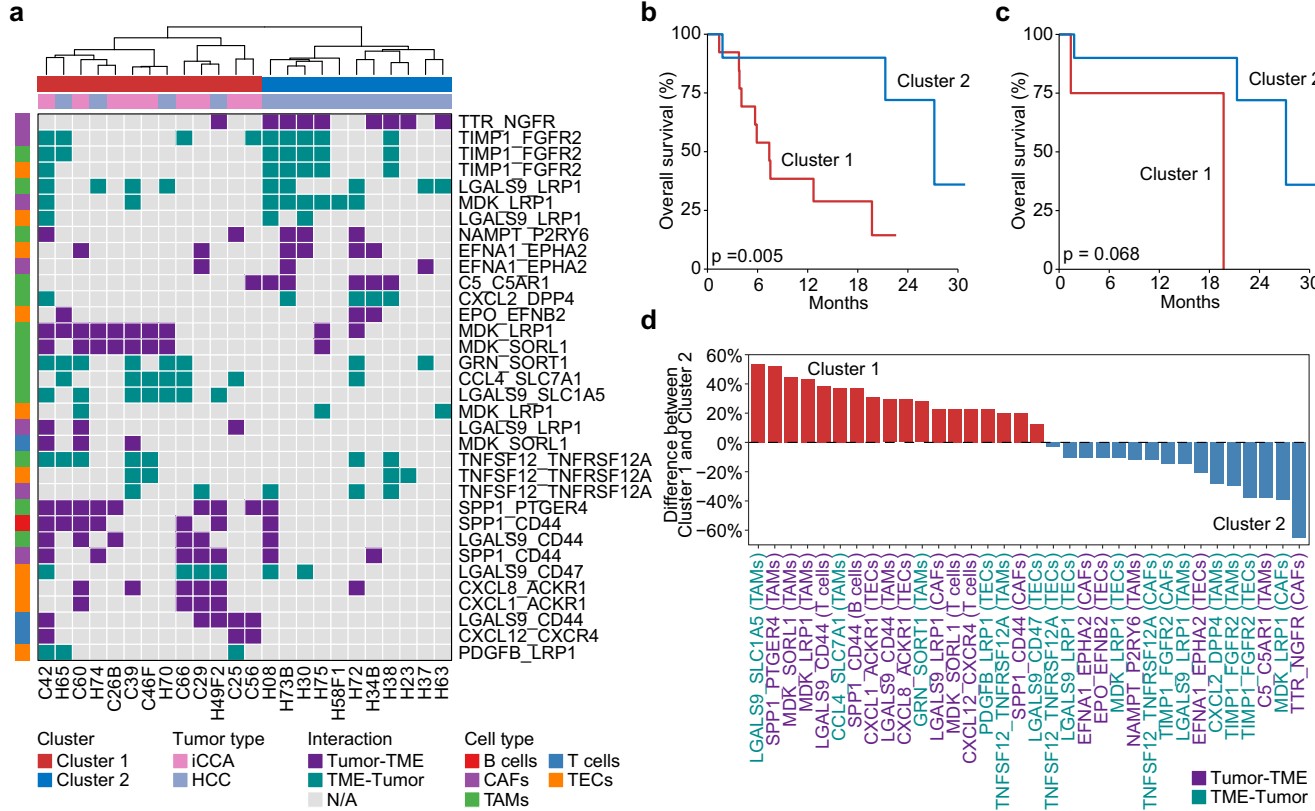

**Fig. 5 | Communication of malignant cells and non-malignant cells are associated with patient outcome. a** Hierarchical clustering of the ligand–receptor interaction patterns of malignant cells and non-malignant cells. Each row indicates a ligand–receptor pair, with the first and the second gene representing a ligand and a receptor, respectively. Each column represents a tumor sample. The direction of an interaction is indicated by color. Purple, malignant cells provide ligands and interact with receptors from non-malignant cells in the TME; green, non-malignant cells in the TME provide ligands and interact with receptors from malignant cells. Distinct non-malignant cell types that interact with malignant cells are indicated by colors. Clusters were determined based on the hierarchical relationship.

**b**, **c** Overall survival of all patients (**b**) or HCC patients (**c**) from Cluster 1 and Cluster 2 in (**a**). Log-rank test was preformed to show the statistical difference of the two groups. **d** The difference between the proportions of each ligand–receptor interaction in Cluster 1 and Cluster 2. Red, pairs enriched in Cluster 1; Blue, pairs enriched in Cluster 2. The direction of each interaction pair is indicated by color. Purple, malignant cells provide ligands and interact with receptors from non-malignant cells in the TME; green, non-malignant cells in the TME provide ligands and interact with receptors from malignant cells. The non-malignant cell types that interact with malignant cells are indicated in parentheses.

we evaluated the difference between the proportions of each ligand–receptor interaction in the two clusters. Noticeably, the ligand–receptor interaction networks are polarized between the two clusters (Fig. 5d). For example, the LGALS9-SLC1A5 and SPP1-PTGER4 pairs, mainly contributed by the communication of malignant cells and TAMs, were among the top interaction pairs from Cluster 1. To confirm the ligand–receptor pairs identified by CellPhoneDB, we further applied CellChat[33] to determine cellular interactions. We found >85% consistency of the ligand–receptor interactions (i.e., SPP1-PTGER4 and

LGALS9-SLC1A5) between tumor cells and TAMs using the two methods, suggesting that the identified pairs are stable (Supplementary Fig. 14a). In addition, we evaluated the impact of the number of malignant cells on ligand–receptor pair identification by randomly sampling malignant cells. We applied this strategy to five cases with the highest number of malignant cells and found no linear relationship between the number of cells and the accuracy of ligand–receptor interaction determination (Supplementary Fig. 14b). However, we did notice a slight drop of its accuracy when the number of malignant cells

is less than 20 (i.e., an average accuracy of 80.04% with 20 malignant cells and 71% with 10 malignant cells (Supplementary Fig. 14b).

To further validate whether the specific ligand–receptor interaction networks were associated with overall survival of HCC patients, we applied the ligand–receptor pairs found from the single-cell analysis to bulk transcriptomic data of 542 patients from three HCC cohorts (i.e., LCI cohort, TCGA HCC cohort, TIGER-LC HCC cohort). Different from directly evaluating cell-cell communications using single-cell data, we developed a strategy by calculating the mean of the ligand and receptor in a specific interaction pair and then determining the occurrence of this pair using the median expression across patients to mimic ligand–receptor interactions among tumor and TME (see "Methods"). We applied this strategy to all the identified ligand–receptor pairs to generate the interaction map of each patient, based on which hierarchical clustering was then performed. We found that patients could be grouped into two main clusters with distinct interaction patterns in the LCI cohort (Fig. 6a) as well as in the TCGA HCC and TIGER-LC HCC cohorts (Supplementary Fig. 15a). To evaluate the consistency of the communication patterns among the three cohorts, we calculated the proportion of each ligand–receptor pair in Cluster 1 for all three cohorts. We found high concordance among three different cohorts with pairs to be assigned to Cluster 1 or Cluster 2, especially for tumor-TAM-derived pairs in Cluster 1, i.e., LGALS9-SLC1A5 and SPP1-PTGER4 pairs (Fig. 6b). Remarkably, patients from the two clusters had a consistently different overall survival in tumor tissues but not in adjacent non-tumor tissues from all three cohorts (Fig. 6c), suggesting that the survival related ligand–receptor interaction activities are imbedded within a tumor lesion. Consistently, when patients from each cohort were divided into three clusters based on the hierarchical relationship, we found a significant trend linking interaction networks to the overall survival of patients, further indicating that the interaction network is a stable feature of HCC aggressiveness (Supplementary Fig. 15b). Collectively, these results imply that the landscape of each tumor ecosystem may contain a unique combination of tumor-immune/stromal interaction pairs from a successful tumor evolution, which is analogous to lock-and-key feature of the enzyme-substrate interaction. These interactions can be exploited as a classifier of tumor aggressiveness.

## Validation of key ligand–receptor pairs by multiplex in situ hybridization in HCC

To further validate the robustness of the ligand–receptor interactions linked to HCC prognosis, we evaluated the top two pairs, i.e., LGALS9-SLC1A5 and SPP1-PTGER4, representing the tumor-TAM communications for proof-of-principle analysis (Fig. 7a), as validation of all significant pairs would be technical challenging. We determined their expression patterns in tissue microarrays consisting of paraffin blocks of both TIGER-LC cohort (HCC, $n = 68$) and LCI cohort (HCC, $n = 190$) using the RNAscope multiplex fluorescent in situ hybridization assay (Fig. 7b). We found a significant correlation of the expression levels of all four target genes in both cohorts between transcriptome-based analysis and RNAscope-based analysis, indicating a good quality of the RNAscope assay of the target genes (Fig. 7c and Supplementary Fig. 16a). Based on the RNAscope signal and the resolved spatial context of each gene, we then determined whether the pair associated genes were more likely to be colocalized than by chance. This was implemented by calculating the Bhattacharyya coefficient (BC) of each tumor, which could measure the amount of overlap of two spatially distributed genes. Specifically, we partitioned each tumor into tiles and calculated the probability of each gene in each tile to calculate the BC for each tumor, with one representing a fully spatial overlap of two genes and zero indicating no overlap (Fig. 7d). In addition, we measured the proportion of the filled tiles (FTs) for each tumor to indicate the spatial preference of the pair-related genes in certain tumor locations (Fig. 7d). We found a high colocalization of the pair-related genes

for both pairs with high BC values, suggesting the co-dependence of the pair-related two genes (Fig. 7e and Supplementary Fig. 16b). Moreover, the pair-related genes were colocalized in certain tumor regions rather than the whole tumor space (Fig. 7f and Supplementary Fig. 16c). Consistent with single-cell and bulk transcriptome data, patients with high expression of both pairs had a significant worse overall survival than those of the low expression groups in both HCC cohorts (Fig. 7g and Supplementary Fig. 16d). Therefore, the co-occurrence of the two ligand–receptor pairs (LGALS9-SLC1A5 and SPP1-PTGER4) was stable and the elevated expressions were associated with poor patient outcome in HCC.

While evidence for physical interactions of LGALS9-SLC1A5 or SPP1-PTGER4 as the ligand and receptor pairs has been described in the curated database of CellPhoneDB[31], the expression patterns of each gene among different cells as well as the functional consequence of these interactions between tumor cells and TAMs have not been determined. We found that SPP1 and SLC1A5 were more abundantly expressed in malignant cells while PTGER4 and LGALS9 were more abundantly expressed in non-malignant cells (Supplementary Fig. 17a). In addition, the expression of these genes was also significantly higher in tumors with the two pairs than those without the two pairs (Supplementary Fig. 17b). To explore the biological features associated with this unique tumor-macrophage link, we compared single-cell transcriptomic profiles of TAMs from tumors with or without the presence of LGALS9-SLC1A5 and SPP1-PTGER4, and found 4 main subtypes of TAMs, i.e., c1 (proliferative), c2 (inflammatory), c3 (restorative), and c4 (CLEC9A$^+$WDFY4$^+$) (Fig. 7h). We found that c1 and c4 TAMs were enriched in tumors with the pairs while c2 and c3 TAMs were enriched in tumors without the pairs (Fig. 7i). We also searched for differentially expressed genes in TAMs or tumor cells from the cases with or without the specific pairs as the biological surrogates of the unique tumor-macrophage link (Fig. 7j and k). TAMs from tumors with the pairs had unique transcriptome profiles with enriched genes involving in oxidative phosphorylation while tumor cells from cases with the pairs showed unique transcriptome profiles with genes enriched in interferon response pathways (Supplementary Data 1). Consistently, both TAM- and tumor cell-derived surrogate signatures could significantly discriminate Cluster-1 HCC from Cluster-2 HCC in all three cohorts evaluated, indicating the biological importance of the surrogate signatures (Supplementary Fig. 17c, d). In addition, there was a significant correlation between the expression of the two ligand–receptor pair-related four genes and their functional surrogate genes (Supplementary Fig. 17e–g). Collectively, these results indicate the presence of a unique and stable tumor-macrophage signaling activity representing unique signatures downstream of the specific ligand–receptor interactions linked to HCC aggressiveness.

## Discussion

Because of tumor evolution and the consequent ITH, cancer research has been confronted for decades by the dilemma as how best to effectively define key drivers and functional biomarkers representing the hallmarks of cancer as the basis for implementing early diagnosis and precision intervention[4]. The approaches by the TCGA and ICGC consortia to provide big data analytics especially integrative genomics are exciting and enable a rich data source for driver discovery. These initiatives also promote the idea of targeting tumors based on drivers unique to certain molecular subtypes as the central theme of precision oncology[34]. While this strategy led to initial success in targeting melanoma with BRAF mutations or lung cancer with EGFR mutations, most tumors eventually relapse and the overall prognosis of those patients remains poor[35,36]. This is especially challenging for liver cancer research in which a complex etiology-related hepatocarcinogenesis results in a vast heterogeneous cancer genome without dominant driver mutations but plenty of passenger mutations that do not provide any phenotypic consequences[37]. Tumor evolution may be the

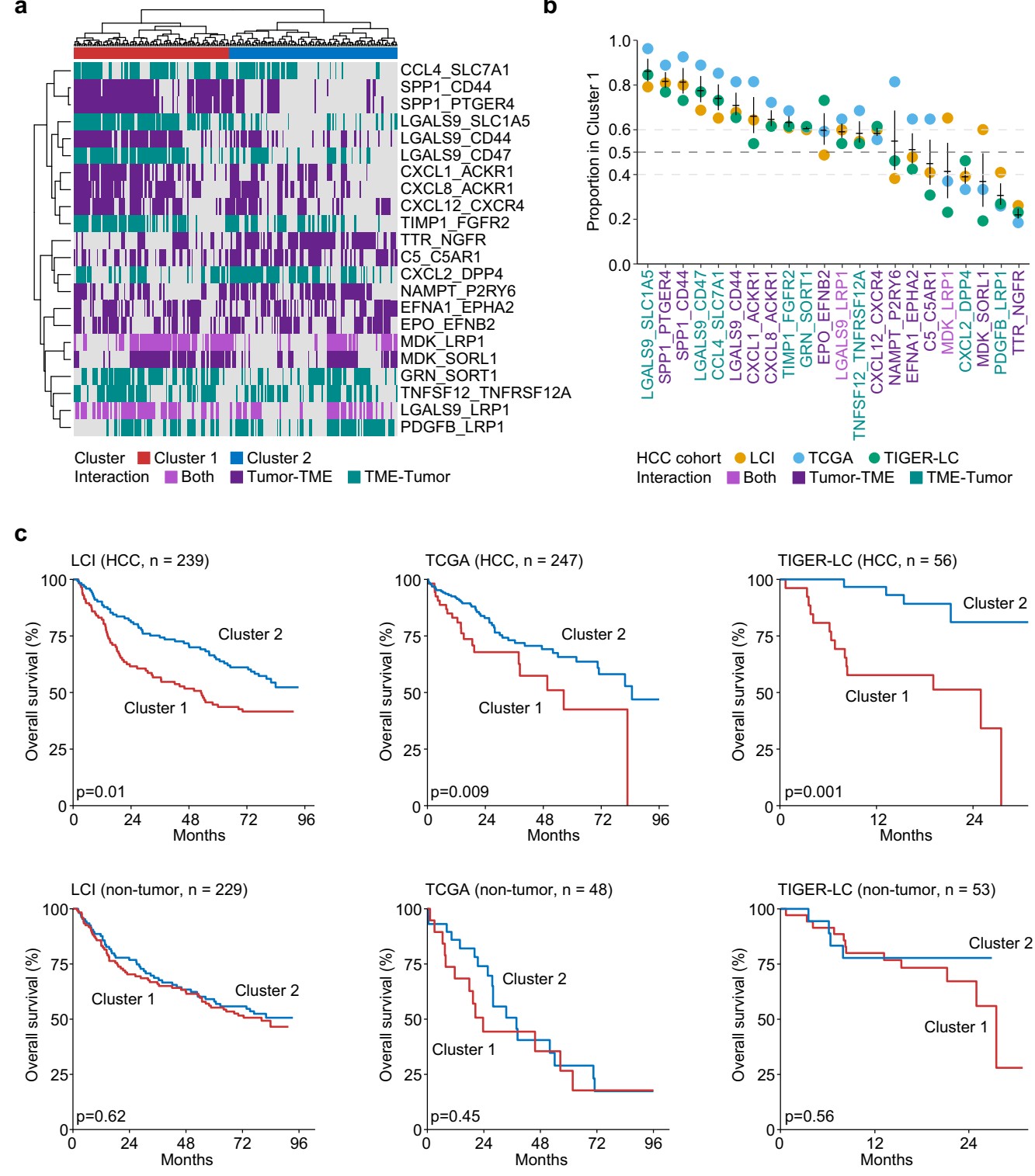

**Fig. 6 | Validation of the tumor and TME interaction patterns for patient stratification using bulk transcriptomic data. a** Hierarchical clustering of ligand−receptor interaction activities in LCI cohort. Each column represents a tumor sample. Each row represents a pair. The direction of each interaction pair is indicated by color. Purple, tumor provides ligands and interacts with receptors from the TME; green, TME provides ligands and interacts with receptors from tumor; light purple, both directions (pairs were identified in both directions from single-cell analysis but can only be modeled once in bulk data). **b** The proportion of each pair in Cluster 1 of three HCC cohorts. Error bar, mean ± standard error of the mean. **c** Overall survival of patients from Cluster 1 and Cluster 2 in three HCC cohorts and the corresponding non-tumor cohorts. Cluster 1 and Cluster 2 were determined based on hierarchical clustering of ligand−receptor interaction activities in (**a**) and Supplementary Fig. 15a. The number of samples in each cohort was provided. Log-rank test was preformed to show the statistical difference of the two groups.

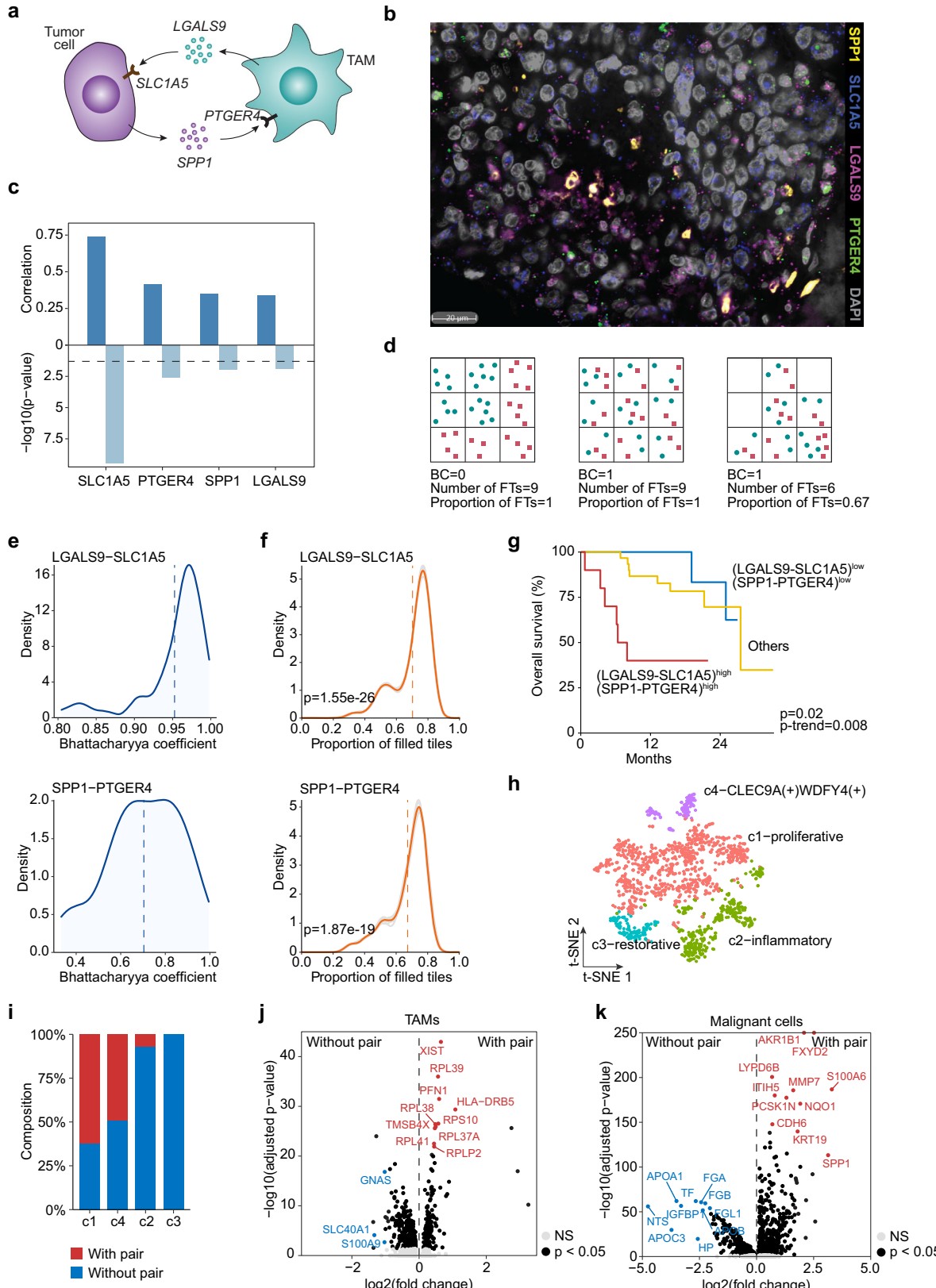

main reason for therapeutic failure and consequent poor patient outcomes. Recent advances in single-cell technologies have provided an unprecedented sensitivity to better define a tumor ecosystem[12]. An important question remains as to what key features represent intrinsic tumor biology and its evolution for each tumor lesion and how much sampling bias contributes to the observed ITH.

The success in establishing a tumor colony should satisfy two parallelly evolving processes, i.e., an acquisition of molecular alterations in somatic cells and an appropriate 'molding' of tumor cell landscape known as the TME to support the survival and fitness of somatically altered cells. Therefore, a perfect fit between tumor cells and their TME may reflect a timestamp for a successful tumor

**Fig. 7 | Validation of two interaction pairs between tumor and TAM using RNAscope assay. a** Illustration of ligand–receptor interaction pairs between tumor cell and TAM (generated using BioRender). **b** A representative image of RNAscope multiplex fluorescent in situ hybridization of four genes of an HCC sample from a total of 258 samples analyzed. **(c)** Correlation of RNAscope signal and bulk transcriptome gene expression in TIGER-LC cohort. Pearson's correlation coefficient (two-sided) was calculated. Dashed line: $p = -\log10(0.05)$. **d** Evaluation of the colocalization of two spatially distributed genes. BC, Bhattacharyya coefficient: 1, a full colocalization; 0, no colocalization. FTs, filled tiles. **e** The distribution of BCs of LGALS9 and SLC1A5 (top) as well as SPP1 and PTGER4 (bottom) in HCC samples from the TIGER-LC cohort. Dashed line: mean value. **f** The distribution of the proportion of filled tiles of LGALS9 and SLC1A5 (top) as well as SPP1 and PTGER4 (bottom) in HCC samples from the TIGER-LC cohort. Ten-times of randomization was used to generate random spread of markers on tissue sections as a reference.

Gray line, proportion determined based on the ratio of true signal and each random spread; gold line, mean derived from ten gray lines. Dashed line: mean value. Student's t-test (two-sided) was applied. **g** Overall survival of HCC patients with low expression and high expression of LGALS9 and SLC1A5 as well as SPP1 and PTGER4 from the TIGER-LC cohort. Tumor samples with expression of the four marker genes in between were grouped into others. Log-rank test and a trend test among the groups were preformed. **h** t-SNE plot of TAMs from samples with or without the two pairs (i.e., LGALS9 and SLC1A5, SPP1 and PTGER4) in Fig. 5a. **i** The composition of each TAM cluster. **j** Differentially expressed genes of TAMs from the samples with or without the two pairs (i.e., LGALS9 and SLC1A5, SPP1 and PTGER4) in Fig. 5a. **k** Differentially expressed genes of malignant cells from the samples with or without the two pairs (i.e., LGALS9 and SLC1A5, SPP1 and PTGER4) in Fig. 5a. Wilcoxon test with multiple test adjustment was applied in (**j**) and (**k**).

evolution and should be unique to each solid tumor lesion. This is analogous to the lock-and-key model to explain the enzyme-substrate interaction in efficiently achieving a biological process. In this study, we attempted to find evidence of a lock-and-key feature in HCC by incorporating spatial single-cell analysis to examine whether sampling bias may contribute to the appearance of ITH and to search for molecular fingerprints of tumor-stromal interactions unique to tumor biology of each liver tumor lesion. We found that some differences in the tumor cell composition and non-malignant cells in the TME could be observed among biopsies from different sampling regions. However, molecular features representing the specific ligand–receptor interactions among tumor cells and non-malignant cells appeared stable among various cohorts from patients with different ethnicities and etiologies. Specifically, we identified two ligand–receptor pairs, i.e., LGALS9-SLC1A5 and SPP1-PTGER4 that may represent a fingerprint of functional interactions between tumor cells and TAMs. We found that SLC1A5 and SPP1 are mainly produced by tumor cells while LGALS9 and PTGER4 are mainly expressed in TAMs. It should be noted that SPP1 is also expressed in TAMs, but the expression is much lower than tumor cells. In addition, SPP1 encodes osteopontin (OPN), a cytokine known to promote HCC metastasis[21]. It is known that OPN exists several different isoforms. However, it is difficult to determine the status of OPN isoforms using 10x genomics single-cell data. This would be an interesting subject of future studies. SLC1A5 encodes a neutral amino acid transporter[38]. In contrast, PTGER4 encodes one of four receptors for prostaglandin E2, which may be involved in T-cell factor signaling. PTGER4-expressing macrophages have been shown to promote intestinal epithelial barrier regeneration upon inflammation[39]. LGALS9 encodes an S-type lectin involved in cell adhesion, immune escape, angiogenesis and tumor metastasis[40]. While physical interactions among LGALS9-SLC1A5 and SPP1-PTGER4 pairs have been described in the curated database of CellPhoneDB[32], a functional consequence of these interactions between tumor cells and TAM has not been determined. We found a significant correlation between differentially expressed genes in tumor cells and TAMs defined by the presence of the fingerprint pairs and the ligand–receptor expressions themselves, suggesting that the activities of these genes may represent a functional interaction of the unique tumor-macrophage network. Moreover, we found that tumor cells with the fingerprint pairs had a unique transcriptome enriching with interferon response pathways, while tumor cells without the fingerprint pairs enriched cellular signaling involving coagulation. In contrast, TAMs with the fingerprint pairs were much more proliferative and enriched genes were involved in oxidative phosphorylation. It is interesting to note that among the identified TAMs, we found a cluster of CLEC9A+WDFY4+ cells. While CLEC9A is a marker for type 1 dendritic cells, it is also expressed in other cell types including macrophages. We further validated the presence of the fingerprint pairs to be associated with overall survival in two independent cohorts of 258 HCC patients using an in situ hybridization

approach. Our results suggest that the transcriptome-based single-cell analysis is a robust tool to define each tumor lesion reflecting its tumor biology. The identification of the unique ligand–receptor fingerprint pairs may help provide the rationale for implementing biopsy-based single-cell analysis for biological understanding of tumors in questions for clinical decision making, which should be evaluated further in clinical trials.

HCC and iCCA are two histological subtypes of liver cancer. Bulk transcriptomic profiling of primary HCC and iCCA indicates both distinct and shared molecular features[41]. In the clustering analysis of ligand–receptor activities using our single-cell cohort, we found Cluster 1 comprised both HCC and iCCA while Cluster 2 was mainly composed of HCC (Fig. 5a), indicating some HCC shared ligand–receptor communication features with iCCA while others did not. Genomic studies of liver cancer demonstrated trunk and branch mutations along with distinct genetic clones among different regions of a liver tumor. However, we found transcriptomic profiles of malignant cells are similar among different sampling locations of a tumor lesion for most of the cases in our single-cell cohort. A recent single-cell study of lung tumor evolution using both scRNA-seq and single-cell DNA sequencing (scDNA-seq) demonstrated that the clones determined by scDNA-seq are largely independent from clones with similar cellular states derived from transcriptomic landscape determined by scRNA-seq[7]. These results indicate that genomic alterations may be independent of transcriptomic profiles. This is anticipated as most genomic alterations used for clonality analysis have no functional consequence as most of them are passengers[1]. Consistently, many recent studies including this study have now shown evidence supporting the idea that cellular states defined by scRNA-seq may be better in representing tumor cell clonality and evolution[20].

One limitation of this study is that single-cell data were based on a small cohort, especially for iCCA samples, as well as limited longitudinal biopsies for monitoring identified fingerprint during tumor evolution. Another limitation is that the validation of the LGALS9-SLC1A5 and SPP1-PTGER4 interactions is not at the protein level. It's very challenging to target different proteins with multiple antibodies on the same slide using multi-channel chromogenic detection. For this reason, our effort was mainly relying on the use of RNAscope analysis as a proof-of-principle experiment even though this approach has its limitation and therefore needs to reach interpretation with caution. In addition, the functional consequences of the LGALS9-SLC1A5 and SPP1-PTGER4 interactions have not been tested experimentally using both in vitro and in vivo HCC models. However, we validated the stability of the identified interaction networks using bulk transcriptomic data of an additional 542 HCC patients, a necessary step to strengthen their pathophysiological relevance. There is an urgent need in identifying appropriate preclinical models used to explore functional relevance of research biopsy-based observations, a call for ensuring rigor and reproducibility of the follow-up functional studies[42]. Moreover, it is worth to determine the changes of the communication networks

between tumor and TME during tumor evolution in response to therapy and this knowledge may help understand the mechanism of therapeutic resistance. We continue to enroll HCC and iCCA patients with on-treatment longitudinal biopsies at the NIH Clinical Center as a part of the NCI-CLARITY study to address this question in the future.

Liver cancer remains one of the most difficult to treat solid malignancies with a 5-year survival rate of less than 18% in the U.S. for many decades[43]. In the single-cell studies of different cancer types including liver cancer, the phenomenon of extensive tumor heterogeneity has been noticed, which creates a major barrier for effective cancer interventions. Sampling bias could be an issue when one uses a single biopsy to determine tumor biology and response to treatment. Thus, in clinical practice, it is important to identify features that are relatively stable and can be used to assess molecular features of a tumor during the course of clinical intervention to avoid sampling bias. Our study indicates that a unique tumor-macrophage link via ligand–receptor interactions from LGALS9-SLC1A5 and/or SPP1-PTGER4 signaling pairs appears a stable molecular feature to define ITH linked to overall survival of patients with HCC. This is consistent with the notion that tumor cells continuously communicate with the tumor microenvironment, defining the molecular map underlining tumor biodiversity may be a key to improve our understanding of tumor heterogeneity and further identifying novel therapeutic targets. We suggest that the identified feature may reflect a successful tumor evolution unique to each tumor. Exploiting methods to disrupt these interactions could constitute a viable therapeutic strategy to target HCC and stop tumor evolution, thereby improving treatment efficacy.

## Methods

### Human sample collection
A total of seven primary liver cancer patients treated at the University Medical Center in Mainz and the NIH Clinical Center in Bethesda, have been enrolled prospectively into this study. Among them, three patients were diagnosed with iCCA and four were diagnosed with HCC. Tumor size for each patient can be found in Supplementary Table 1. All patients received surgical resection. A total of five samples from the tumor core, tumor border, and adjacent non-tumor tissue were collected for each patient. Specifically, we collected three samples from the tumor core that were not adjacent, one sample from the tumor border, and one sample from the adjacent non-tumor tissue that was not locally close to the tumor. Each sample was measured about 5 mm diameter in size before single-cell library preparation. Sample collection was performed with written informed consent from patients. We removed one sample (T3) from patient 3C due to single-cell library failure and thus a total of 34 samples were included in this study. This study was approved by the ethics committee of the University Medical Center in Mainz and the National Institutes of Health.

### Single-cell suspension preparation
Resected samples were collected in RPMI 1640 media and were immediately processed to keep ischemic time to a minimum. Samples were minced in petri dishes on ice and transferred into gentleMACS C Tubes. Enzymes of the Miltenyi tumor dissociation kit had been added before according to the kit user guide. The tube was placed into the MACS Dissociator for mechanical dissociation. The tube was incubated for 30 min at 37 °C under continuous shaking. Next, the cell suspension was filtered through a nylon mesh and cells were counted to determine number of cells and viability. The samples were then centrifuged at $300 \times g$, 4 °C for 5 min and re-suspended in freezing media for cryopreservation in liquid nitrogen. Samples from Mainz, Germany, were shipped cryopreserved to the NIH.

### Single-cell library preparation and droplet-based scRNA-seq
Cryopreserved samples were thawed and prepared according to the Single Cell 5′ Reagent Kits User Guide. Specifically, the samples were

washed and re-suspended in PBS + 0.04% BSA. Cell viability was determined. The cDNAs were obtained after the GEM (Gel Bead-in-emulsion) generation and barcoding, followed by GEM RT (reverse transcription) reaction. Purified cDNA was amplified for 14 cycles. A clean up using SPRIselect beads was performed. cDNA libraries were prepared with 10x Genomics Single Cell 5′ library & gel bead kit v1.1. cDNA concentration was determined by Bioanalyzer. Libraries were then pooled and normalized to a final loading concentration. The samples were loaded in the lanes according to the 10x Genomics 5′ User Guide and were then sequenced using Illumina NovaSeq platform at Frederick National Laboratory for Cancer Research Sequencing Facility, with sequencing parameters of 28 bp (Read1), 8 bp (Index1), and 98 bp (Read2). The targeted sequencing depth for each sample is 50,000 reads/cell. Base calling was carried out with Real-Time Analysis software (version 3.4.4) on Illumina sequencing systems. Demultiplexing was then performed by using bcl2fastq (version 2.20), with one mismatch allowed in the barcodes. The standard 10x Genomics Cell-Ranger (version 3.1.0) pipeline was used to extract FASTQ files and to perform data processing including alignment, tagging, gene, and transcript counting. Sequenced reads were aligned to human reference sequence (refdata-cellranger-GRCh38-3.0.0) provided by the 10x Genomics.

### scRNA-seq data pre-processing
We integrated single-cell profiles from different samples by performing read depth normalization for all the 34 samples using *cell-ranger aggr* pipeline from the Cell Ranger (version 3.1.0), which equalized the average read depth per cell between samples based on the confidently mapped reads. R Seurat package (version 3.1.2) was applied for the pre-processing of the aggregated data. We kept genes that were expressed in at least three cells and removed cells with less than 500 genes detected. A total of 112,506 cells passed this initial quality control, with an average of 1067 genes and 3359 unique UMIs detected per cell. We then normalized the total counts in each individual cell to 10,000, followed by log transformation to generate the normalized data.

### Separation of malignant cells and non-malignant cells
To separate malignant cells and non-malignant cells, we inferred large-scale chromosomal copy-number variations (CNVs) based on single-cell transcriptome profiles as described in previous published single-cell studies[17,20,44,45]. Briefly, this method infers CNVs by taking the average expression of a set of genomically adjacent genes along each chromosome to eliminate gene-specific patterns and yield CNV profiles, with the assumption of aberrant karyotypes in malignant cells. Because adjacent non-tumor tissues are available in this study, we used cells derived from those samples as a reference during CNV inference in order to reduce background noise. From the inferred CNVs, gains on chromosomes 1q and 8q of malignant cells were observed (Supplementary Fig. 1a), consistent with the CNV profiles generated from genome sequencing of liver cancer[41]. In contrast, no obvious CNV patterns were observed in non-malignant cells (Supplementary Fig. 1a). To further confirm the successful separation of malignant cells and non-malignant cells, we evaluated the expression of epithelial- and liver-specific marker genes in the derived malignant cells and non-malignant cells, considering the epithelial origins of malignant cells[17]. We found consistency of tumor aberrant karyotypes and the marker gene expression, with strikingly higher expression of epithelial- and liver-specific genes in malignant cells, further suggesting a confident separation of malignant cells and non-malignant cells (Supplementary Figs. 1b and 1c). As expected, a high number of genes was expressed in malignant cells, with an average of 3106 genes and 16,658 UMIs detected per malignant cell. In the downstream analysis of malignant cells, we only included the samples with >10 malignant cells detected. With this criterion, we didn't

detect enough malignant cells in 2C as well as the samples labeled as N.D. (not detected) in Fig. 1f.

## Tumor heterogeneity

To evaluate tumor heterogeneity of malignant cells, we used three independent approaches: (1) Dimensional reduction algorithm. We first applied principal component analysis (PCA) to the top 2000 most variable genes of all malignant cells determined using standardized means and variances. We further performed dimensional reduction on the first 20 principal components (PCs) by employing t-distributed stochastic neighbor embedding (t-SNE) method. Samples with more than 10 malignant cells detected were involved in this analysis. In the two-dimensional t-SNE space, we observed heterogeneous tumor cell populations, with a larger tumor heterogeneity between patients than within a patient. (2) Hierarchical clustering method. We also generated a hierarchical tree of malignant cells from different tumor regions of all the cases. Here, we applied the top 2000 most variable genes as described above and calculated the mean expression of each individual gene in all the malignant cells of each tumor sample. Then we constructed the hierarchical tree of all the tumor samples by using correlation-based distance measurement and ward.D2 data agglomeration method. (3) Pearson's correlation analysis. To quantitively measure the level of intraregional (within a tumor region), interregional (across multiple tumor regions within a case) and intertumoral (across multiple tumors from different cases) heterogeneity, we calculated the correlation of the malignant-cell transcriptomic profiles based on the top 2000 most variable genes as described above. Intraregional tumor heterogeneity was measured as the distribution of the pair-wise correlation of malignant cells within a specific tumor region. Interregional tumor heterogeneity was determined as the density distribution of the correlation coefficients of malignant cells across different tumor regions within a specific case. Intertumoral heterogeneity was calculated as the correlation of malignant cells among the samples from different cases. For all the three levels of heterogeneity, Pearson's correlation coefficient was applied, with 1 indicating a perfect positive linear correlation, −1 representing a perfect negative correlation and 0 standing for no linear correlation.

## Cellular dynamics determined by RNA velocity method

To recover the cellular dynamics of malignant cells, we applied RNA velocity method from the scvelo python package. RNA velocity allows for learning of cellular lineage trajectory by considering the spliced and unspliced events in the single cells[23,24]. Specifically, we used the *scv.tl.recover_dynamics* function to recover the full dynamics, followed by calculating the velocity of each gene using *scv.tl.velocity* function based on the splicing kinetics. Then the collection of the velocities of all genes were used to calculate the transition probabilities between cells and to predict the direction and movement of a specific cell in a future state. Finally, the velocities were projected onto a uniform manifold approximation and projection (UMAP) embedding by using the *scv.pl.velocity_embedding_stream* function for visualization. Based on the transcriptional dynamics, the latent time underlying cellular processes of each individual cell was recovered using *scv.tl.latent_time* function. Cases with more than 50 viable malignant cells were used for RNA velocity analysis since it failed to recover the full dynamics of malignant cells if the number of cells is too small. To indicate gene expression along cellular latent time determined by RNA velocity analysis, we applied 10 genes (tumor stemness-related genes and tumor evolution-related genes) including EPCAM, KRT19, ICAM1, PROM1, LGR5, CD44, ANPEP, HNF4A, ALDH1A1, and SPP1 in our analysis[16,20]. Due to varying expression of these genes in the malignant cells of each case, we only included the genes that were expressed in at least 20 malignant cells using the function *scv.pp.filter_and_normalize* with parameters of min_shared_counts=20 and n_top_genes=2000 in the latent time analysis.

## Analysis of non-malignant cells

We first performed PCA of non-malignant cells based on the top 2000 most variable genes determined using standardized means and variances. Then we applied UMAP analysis to the first 20 PCs for dimensional reduction and data visualization. We determined cell lineage of non-malignant cells based on the expression of lineage-specific marker genes to T cells, B cells, TAMs, CAFs, TECs, hepatocytes, and cholangiocytes. To ensure a pure population of T cells, we evaluated the total expression of CD3D, CD3E, and CD3G in each individual cell within the T-cell group, and further calculated the proportion of T cells positive for the three marker genes in each T-cell cluster determined using the Louvain algorithm, where the clusters with a proportion of <80% were removed. To measure the similarity of non-malignant cells from different tumor regions within each case, we calculated the correlation of each cell type between different regions. We first selected most variable genes of each cell type, and then applied the mean expression of the determined most variable genes of each tumor region within a case for correlation analysis (Fig. 3e and Supplementary Fig. 9a). In addition, for each cell type, we calculated the pair-wise correlation of cells within a tumor region to represent the intraregional heterogeneity, and then the average correlation value of each region was applied to measure the ratio of tumor border and tumor core (Supplementary Fig. 9b).

## Immune features of the TME

We generated pseudo-bulk data based on the single-cell profiles of each case and applied the immune signature[46], cytotoxic and exhaustion signatures[47,48] to determine the overall immune scores of the TME by using the average expression of the genes from each signature. In addition, we calculated the ratio between cytotoxic and exhaustion immune scores to reflect the overall immune microenvironment of each case.

## Communication of malignant cells and the TMEs

We resolved the communications between malignant cells and the TMEs by identifying ligand–receptor interactions using CellPhoneDB[31,32], which predicts the significance of a ligand–receptor pair based on the mean expression of the ligand and receptor in two evaluated cell types while using random permutations as background signals. The original significant ligand–receptor pairs were determined with *p* value <0.05 to indicate their enrichment in the interacting pairs of cell types (Supplementary Fig. 11). We used all cells within a case for the original search and further evaluated the derived pairs in each tumor region. In this study, we applied more stringent criteria of *p* value <0.01 and mean expression of a ligand-receptor pair >0.5, in order to find biologically more meaningful pairs. We also removed the ligand-receptor pairs which occurred in the same interacting partners of all patients, given the reason that those pairs represent the common features in all patients and may not contribute to the patient-specific communication networks between the tumor and the TMEs. The derived ligand-receptor pairs were shown in Fig. 4a. Based on the ligand-receptor pairs identified in each individual case, we calculated regional stability. Specifically, we determined the proportion of the pairs that were found in all tumor regions or part of the tumor regions. We also determined the interactions between malignant cells and T-cell subtypes using similar strategies. Because the number of cells varies among cell types, we also evaluated the influence of the number of cells on the ligand-receptor interaction search. We performed random selection of T cells and TAMs, which represent two largest populations of non-malignant cells in our study. We then searched for ligand-receptor interactions based on the randomly selected cells. We repeated the process of random selection and interaction pair search for 10 times and found that an average of 92% ligand-receptor pairs can be matched to the original search, suggesting that the identified

ligand-receptor pairs are very stable (Supplementary Fig. 13). To confirm our findings of the ligand-receptor interactions (i.e., SPP1-PTGER4 and LGALS9-SLC1A5) between tumor cells and TAMs, we applied CellChat[33] to further identify cellular communications. Since the two interactions pairs are not listed in the CellChat database, we manually curated the two pairs. With the derived significant interaction pairs, we also applied a stringent criterion with a probability ≥0.01 similar to our analysis using CellPhoneDB to identify biologically more meaningful pairs.

## Interaction networks of tumor and the TMEs for patient stratification

We used our NIH single-cell cohort to increase the power of studying the communications of malignant cells and the TMEs in liver cancer[20]. In this cohort, single-cell transcriptomes are available for a total of 46 tumor samples from 37 HCC or iCCA patients. The tumor samples with viable malignant cells were used for modeling. For patients with longitudinal tumor samples available, we applied the first sample with viable malignant cells in our analysis. We first searched for ligand-receptor interaction pairs between tumor cells and stromal/immune cells using CellPhoneDB[31,32] and identified the pairs based on the criteria of $p$ value <0.01 and mean expression of ligand-receptor pair >0.5, as used in our multiregional single-cell analysis. We then filtered the derived interaction pairs based on those found in the multiregional analysis, as those pairs have been selected using stringent criteria and demonstrated stable across multiple tumor regions. We also removed the interaction pairs that were occurred in less than three cases or more than ten cases, the same idea of selecting most variable genes for downstream data analysis considering their functional importance. Hierarchical clustering of the interaction network was performed using Pearson's correlation-based distance measurement and ward.D2 data agglomeration method. Two clusters of tumors were derived with different interaction patterns and were associated with distinct patient outcomes. In addition, we evaluated the impact of the number of malignant cells on ligand-receptor pair identification by randomly sampling malignant cells. We selected five cases with the highest number of malignant cells in Fig. 5a and randomly sampled 200, 100, 50, 20, and 10 malignant cells from each case for ligand-receptor pair identification. All the non-malignant cells were used in the analyses. We performed five times of random sampling for each setting and further compared the identified pairs with those in Fig. 5a to determine the accuracy of ligand-receptor pair detection.

To further evaluate the communication networks for patient stratification, we analyzed bulk transcriptomic data from three HCC cohorts including both tumor and paired non-tumor samples, i.e., LCI cohort (tumor, $n = 239$; non-tumor, $n = 229$), TCGA HCC cohort (tumor, $n = 247$, non-tumor, $n = 48$), and TIGER-LC HCC cohort (tumor, $n = 56$; non-tumor, $n = 53$). As we cannot directly measure cell-cell interactions using bulk transcriptome, we developed a strategy to evaluate the communication networks for tumor samples. Specially, we first calculated the mean expression of the ligand and receptor from an interaction pair for each individual tumor sample. Then, based on the median expression of the interaction pair across all tumor samples in a cohort, we determined the occurrence of a pair by comparing the expression value in each patient with that median value. Finally, we performed hierarchical clustering analysis of the communication networks using Pearson's correlation-based distance measurement and ward.D2 data agglomeration method. Clusters were determined using *cutree* function in R dendextend package. In the analysis of bulk transcriptome data, we used the same set of ligand-receptor pairs as those in the analysis of single-cell data (Fig. 5a). But the same ligand-receptor interaction pair in different pairs of interacting cell types from single-cell analysis was considered as one pair in bulk data, because cell types cannot be resolved in bulk transcriptome.

We performed the same analysis for non-tumor samples in the three HCC cohorts as a control.

## Single-molecule RNA in situ hybridization on tissue microarrays of HCC

Tissue microarrays (TMAs) were constructed using 1.0 millimeter (mm) cores from formalin fixed, paraffin embedded (FFPE) tissue for LCI and TIGER-LC HCC cohorts[41]. Tissues were mounted as TMAs with Superfrost PLUS Slides (Thermo Fisher Scientific, Cat # 5951PLUS). RNAScope™ fluorescent 4-plex in situ hybridization[49] was performed, by Advanced Cell Diagnostics (acdbio.com), for four genes on 5 μM TMA sections. Specially, paired double-Z oligonucleotide probes were designed against target RNA for LGALS9 (Cat# 1039898-C1), PTGER4 (Cat# 406778-C2), SPP1 (Cat# 420108-C3), and SLC1A5 (Cat# 427588-C4). The RNAscope LS Fluorescent Multiplex Reagent Kit (Cat# 322800) (Advanced Cell Diagnostics) was used with modified pretreatment conditions. FFPE human TMAs were incubated with Leica BOND Epitope Retrieval Solution 2 (ER2) at 95 °C for 20 min. Then RNAscope 2.5 LS Protease III was used for 15 min at 40 °C. Pretreatment conditions were optimized with the RNAscope LS 2.5 Hs-4-plex Positive Control Probe (Cat# 321808), specific to the housekeeping genes of POLR2A, PPIB, UBC, and HPRT1. Negative control background was evaluated using the RNAscope 4-plex LS Multiplex Negative Control Probe (Cat# 321838) specific to the bacterial *dapB* gene. A 3D Histech Panoramic Scan Digital Slide Scanner microscope with a 40x objective was used to generate fluorescent images. Images were analyzed using HALO® Image Analysis Platform (Indica Labs), where single-cell probe copies were quantified with resolved spatial coordinates. Samples with minimum tissues on the TMAs were excluded from downstream data analysis.

## Colocalization of genes

To evaluate the spatial colocalization of a ligand–receptor pair-related two genes in each tumor core from the RNAscope assay, we calculated the Bhattacharyya coefficient (BC) of each tumor, which provides a measurement of the amount of overlap for two spatially distributed genes. Specifically, we generated partitions of each tumor core with tiles of 500 × 500 pixels (-70 × 70 μm). Then the probability of each gene in each tile was calculated. With the derived probabilities, we calculated the BC as:

$$BC(\mathbf{p1}, \mathbf{p2}) = \sum_{i=1}^{n} \sqrt{p1_i p2_i} \qquad (1)$$

where $p1_i$ and $p2_i$ represent the probability of the first and the second gene in the $i$th tile respectively, and $n$ represents the total number of tiles from the partition of the space of a tumor sample. Because probabilities were applied for BC calculation, the derived BC value is between 0 and 1, with 1 representing a full overlap and 0 indicating no overlap. Samples with >1% cells positive for each of the two markers were used for this analysis. To measure the proportion of maker-positive tiles in a tumor sample, we randomly shuffled gene expression of the cells and counted the total number of filled tiles in a random situation, which was used as the denominator for evaluating the proportion of the original resolved marker-positive tiles. This process was repeated for 10 times for each tumor sample as a way of evaluating the spatial preference of a ligand-receptor pair-related two genes.

## Analysis of the interaction pair-related downstream signaling

To evaluate the differences of malignant cells between tumors with the two pairs of LGALS9-SLC1A5 and SPP1-PTGER4 and those without the two pairs, we performed differential gene expression analysis of malignant cells between the two groups of tumors from our NIH

single-cell cohort (Fig. 5a). With the derived genes, we performed gene set enrichment analysis using GSEA (version 4.0.3) to search for enriched pathways in each group. We further selected genes that were highly expressed in the with-pair group (gene set 1, average log[fold-change] > 1 and adjusted $p$ value < 0.05) and the without-pair group (gene set 2, average log[fold-change] < −1 and adjusted $p$ value < 0.05). The two sets of genes were then applied for PCA of bulk transcriptome data in LCI, TCGA HCC, and TIGER-LC HCC cohorts to demonstrate their roles as downstream surrogates of the two ligand-receptor interaction pairs. To further indicate the relationship of the two pairs and the two surrogate gene sets, we calculated the correlation between the geometric mean of the four genes (LGALS9, SLC1A5, SPP1, and PTGER4) and the ratio of the average gene expression of gene set 1 and gene set 2 for all the three HCC cohorts. We did similar analyses for TAMs to demonstrate the two interaction pairs-related downstream signaling in these cells.

### Flowcytometry

Cryopreserved single cells were washed once and re-suspended in Stain Buffer (BD Pharmingen®, Cat. #554656). Cells were then stained with antibodies. For EpCAM staining, cells were stained using PE-conjugated anti-EpCAM antibody (Miltenyi Biotec, Cat.# 130-113-826) or its isotype control (Miltenyi Biotec, Cat. #130-113-762). For GPC3 staining, cells were stained indirectly with mouse monoclonal anti-GPC3 antibody[50] (clone YP7) or its isotype control (BD Pharmingen®, Cat. #555746) together with APC-conjugated goat anti-mouse antibody (BD Pharmingen®, Cat. #550826), respectively. Antibodies were used at 1:100 dilution. Antibody-stained cells were washed with the Stain Buffer and then analyzed with BD FACSCanto II cell analyzer (Becton Dickinson).

### Immunohistochemistry

Deparaffinization and rehydration were performed using xylene and citrate buffer ph 6.0 followed by distilled aqua. Washing steps with TBS-TX 0.03% and Trtion X-100 0.3% TBS were performed before and after incubation with $H_2O_2$. Next, blocking was done using 2.5% horse serum and 0.1% triton over 45 min followed by avidin/biotin blocking (Vectorlabs). Slides were then incubated with mouse anti-CD3 (monoclonal, Santa Cruz, Cat. #sc-59010, dilution 1:80) antibody overnight at 4 degrees Celsius. Next day, several washing steps were performed using TBS-TX, $H_2O_2$, and distilled water. Secondary antibodies using anti-mouse-biotin (streptavidin HRP, Rockland/Thermos Scientific, Cat. #N100, dilution 1:1000) were incubated for 90 min at room temperature followed by several washing steps. Streptavidin-HRP and later DAB (Vectorlabs) were each added in a separate step followed by washing steps. Finally, another hematoxylin staining was performed over 20 s. To quantitively determine CD3+ cells in different tumor regions of a patient, we counted the number of CD3+ positive cells per selected field of view (~0.2 mm). Five fields of view were randomly selected for each tumor region. This was performed for patient 1C, 2C, 3C, 2H, and 3H with tumor blocks available. For SPP1 staining, Osteopontin (OSP)/ SPP1 monoclonal antibody [Clone OSP/4589] (Rat, MyBioSource, Cat. #MBS4382252) was used at 1:100 dilution.

### Histopathology

Paraffin sections were dried overnight on a heating plate at 40 °C. Deparaffinization and rehydration were performed using xylene and EtOH. Sections were first stained in hematoxylin for 3 min and were rinsed in water for three times. Then sections were stained in eosin for 2 min, followed by several dehydration steps starting with 70% EtOH and eventually 100% EtOH. Finally, sections were incubated in xylene for 5 min and covered with cover slides.

### Plot generation

Violin plots, box plots, scatter plots, bar plots, and density plots were generated with *ggplot* and *geom_violin*, *geom_boxplot*, *geom_point*, *geom_bar*, *geom_density* functions in R ggplot2 package (version 3.3.5). Heatmap was generated using *Heatmap* function in R Complex-Heatmap package (version 2.2.0). Kaplan–Meier survival plots were generated with GraphPad Prism (version 8.4.3).

### Reporting summary

Further information on research design is available in the Nature Portfolio Reporting Summary linked to this article.

## Data availability

The processed single-cell transcriptomic data generated in this study have been deposited in the Gene Expression Omnibus (GEO) and are available without restriction under accession number GSE189903. However, the NCI raw sequencing data are considered protected information and access to raw data is therefore restricted. The raw sequencing data are available in the NCBI dbGaP archive under accession number phs003117.v1.p1. Access via the NCI's dbGaP can be requested by qualified senior and principal investigators overseeing the research. The NCI's Data Access Committee reviews such requests and will make data available for up to 12 months. The publicly available datasets used in this study include a processed single-cell data of GSE151530, bulk transcriptomic data of GSE14520 and GSE76297, as well as the TCGA database (TCGA-LIHC). Source data are provided as a Source data file. Source data are provided with this paper.

## Code availability

Code is available upon request. It should be directed to and will be fulfilled by the Lead Contact, Xin Wei Wang (xw3u@nih.gov).

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

## Acknowledgements

We thank members of the Wang laboratory for critical discussions, the patients, families, and nurses for contribution to this study. We also thank Eytan Ruppin and Snorri Thorgeirsson for their critical evaluation of the manuscript. This work was supported by grants (Z01 BC 010877, Z01 BC 010876, Z01 BC 010313, and ZIA BC 011870) from the intramural research program of the Center for Cancer Research, National Cancer Institute of the United States. J.U.M. received funding from the Volkswagen Foundation (Lichtenberg Program) and the Wilhelm-Sander Foundation (2021.089.1).

## Author contributions

L.M. and X.W.W. developed study concept; L.M. performed data analysis; S.H., L.W., F.K., S.K., M.F., S.M.H., A.S., J.M.H., D.M., R.K., T.F.G, J.C., M.R., and J.M. performed sample collection and processing; S.K. and M.K. performed additional analyses; L.M. and X.W.W. interpreted data; L.M., S.H., and X.W.W. wrote the manuscript. All authors read, edited, and approved the manuscript.

## Funding

## Competing interests

The authors declare no competing interests.
