## [Peer review file · Nature Communications]

REVIEWER COMMENTS

Reviewer #1 (Remarks to the Author): Expert in iCCA and HCC genomics, single-cell genomics, and tumour microenvironment

In this manuscript, Ma, et al. performed a multiregional single-cell transcriptomic analysis of HCC and iCCA from seven liver cancer patients. They validated the results in single-cell data from an additional 37 HCC and iCCA patients, as well as several independent cohorts. An important finding of this paper is that specific molecular network of tumor cells and macrophages (i.e. LGALS9- SLC1A5 and SPP1-PTGER4) represent a stable hub of HCC malignancy. In general, this manuscript would be of interest to the growing crowd of oncologists and those interested in single-cell analysis. However, the majority of the work is descriptive and the lack of validation assays to confirm the annotations dampen the enthusiasm, I raise hereafter would potentially increase its impact.

Major concerns:

1. Given the author performed the multi-regional sampling of tumors, it would be interesting to profile the exome profile or computationally infer the copy number to check out the genomic features of tumor subregions.
2. Most of the findings of this study are exploratory and lack experimental verification. It is necessary to confirm and validate key findings based on the scRNA-seq discovery e.g. T-cell composition appears stable among different sampling regions or whether LGALS9-SLC1A5 and SPP1-PTGER4 interactions synergistically promote tumor progression. This is very important for the reliability of the findings.
3. In Figure 3, T-cell composition appears stable among different sampling regions, it would be better to further subdivide T cells into different subsets (e.g. Treg, Tex, etc.) and reveal their differences in distribution, functional status, and interactions between tumor border and core regions.
4. The unsupervised clustering analysis and RNA velocity analysis showed that malignant cells were similar among different tumor regions. Those data were partly inconsistent with previous findings (Ann Oncol. 2019 Jun 1;30(6):990-997.; Gastroenterology. 2022 Jan;162(1):238-252). The authors should provide more evidence to re-confirm their findings and discuss the potential explanations.
5. The number of tumor cells obtained from different regions is too small, which will reduce the accuracy of subsequent studies on the interaction between tumor cells and TME, as well as the issue

raised above. For example, 3H and 2H contains only tens of cancer cells. It would be better if the authors utilized FACS or computational strategies to solve this problem.

6. It has been reported that not only tumor cells, but also a subset of macrophages highly express SPP1 (Cell. 2021 Feb 4;184(3):792-809.e23.). Also, two isoforms of SPP1: a secreted form and an intracellular form has been identified with distinct functions (Immunol Res. 2011 Apr;49(1-3):160-72.). So, it is difficult to reveal the true interaction between tumor cells and TAMs through SPP1-PTGER4 simply relying on scRNA-seq data analysis. This should be described more detail to avoid misunderstanding.

7. What are the overall features of the TME of enrolled patients (i.e. immunosuppressive or immune activated)? Also, the annotation of immune cell subsets is required in Fig. 3a (e.g. Tex, Tem, and etc.). Given the large difference of phenotypic and molecular difference of T cell subsets, it would be better to analyze the specific cell-cell interaction between tumor and T cell subsets in Fig 4a.

8. Inferring the cell-cell interaction only based on CellPhoneDB is not enough. I suggest the authors to repeat the analysis by using other algorithms (e.g. CellChat) to reconfirm their findings.

Minor concerns:

1. The process of sample acquisition should be described in more detail, such as the size of tumor pieces, the distance between tumor border and core, and whether the three regions of tumor core are adjacent.
2. The authors should specify the count of UMIs/genes of detected cancer cells.
3. Not all the tumor samples and regions were enrolled for the scRNA-seq analysis (2C, 1CT2, etc.), Please explain this in the main text.
4. The authors should provide more technical description on the data integration of single cells from different patients.
5. The authors should discuss the potential clinical impact of their paper.

Reviewer #2 (Remarks to the Author): Expert in hepatocellular carcinoma tumour immune microenvironment and immunology

The manuscript by Ma et al. performed multiregional single-cell transcriptome profiling of 7 liver cancer patients (4 HCC and 3 iCCA patients), validated using single-cell seq data in additional 37 HCC and iCCA patients and identified specific molecular network/fingerprint consisting of LGALS9-SLC1A5, SPP1-PTGER4 as tumor and macrophage-derived ligand-receptor interaction pairs, linked to tumor aggressiveness. Independent validation of these interaction pairs was performed using bulk transcriptome profiles of 542 HCC samples from 3 independent cohorts and multiplexed fluorescence in situ hybridization-based profiles of 258 HCC samples from 2 cohorts. The study consists of deep and comprehensive immunoprofiling at single-cell levels and large validation datasets, showing important findings on a stable molecular network that could be targeted therapeutically. However, in order to support the above claims, there are multiple areas where data presented will require enhanced description and explanation. Also, some further validation data could be provided.

These are the major comments:

1. Spatial single-cell analysis is typically linked to spatial transcriptomic analysis with matched histochemistry data and since the authors have in fact performed “multiregional single-cell transcriptomic profiling” on 7 liver cancer patients (4 HCC and 3 iCCA patients), while majority of the validation were performed on single-cell data (37 liver cancer patients) and other databases, I would recommend to rephrase the title of the manuscript e.g. to remove the word “Spatial”, to more appropriately reflect the main findings of the current study.
2. Fig. 1, why is the data from sample 2C missing? In that case, should the number of samples be revised to only 6 liver cancer patients (4HCC + 2 iCCA)? Also it seems like not all patients consistently have data from 3 tumour cores T1-T3 and 1 tumor border (B), this should also be indicated clearly in the results. Next, are each dataset normalized by its respective Normal tissues as control? It will be good to show all the data with Normal vs Tumours vs Border as Normal would be expected to be distinctly different from tumours. It is not known how different Tumour Cores vs Border should be distinctive, hence it will be good to check if N is actually separated from T.
3. It is not clear how heterogeneity scoring is determined, based on RNA expression of each single cell? All genes or selected genes? This information is not immediately clear from the description which makes interpretation of comparison between inter- or intra-tumoural heterogeneity difficult.
4. Are there any distinctive differences between HCC vs iCCA?
5. Fig. 2 showed only one sample 1C while Fig. S3 showed another 3 (3C, 1H and 4H), in fact data from all samples should be shown. The trend are in fact not very obvious for other cases and it is not clear why not the same genes were shown in Fig. 2 vs Fig. S3.
6. Could the lower intratumoural heterogeneity of T cells due to more equal physical distribution within the TME where the other cell types may show more heterogenous distribution in TME hence leading to higher heterogeneity? In another word, is the heterogeneity physical or genetic/phenotypic? As the stable molecular feature is the key focus of the study, it is essential to determine this e.g. By IHC analysis of each of these subsets in the TME.

7. If T cells are the most stable subset within the TME, why is the subsequent stable ligand/receptor pairs were identified between malignant/TAM instead? How stable and heterogeneity of TAM in TME?

8. Fig. 4C, how is regional stability calculated? Data shown in Fig.4E & 4F is interesting, though most likely due to a more stable interpatient TME compared to malignant cells (Fig. 1 vs Fig. 3).

9. SPP1 is well known to be macrophages-specific genes however according to their CellPhoneDB data, it is expressed on tumour binding to PTFER4 expressed on TAMs. In my opinion, it is important to determine where exactly is SPP1 expressed in HCC tumours e.g. by IHC.

10. Validation shown in Fig. 7 should be done using IHC instead to show the actual protein expression and ligand/receptor interaction. Also, no tumour or TAM markers were used together to specify the sources of these markers.

Minor comment:

1. I have noticed some discrepancies in number of cases or samples involved in the data presented, please check to verify the accuracy of reported numbers or to explain why if any data have been omitted.

Reviewer #3 (Remarks to the Author): Expert in single-cell RNA-seq analysis

In this manuscript, the authors aim to characterize the characteristics and intra-tumor and inter-tumor heterogeneity using single-cell RNA-seq profiling of multiple spatial sites of the same tumor across multiple tumors in hepatocellular carcinoma. Analysis of tumor cells demonstrates that the tumor cells from the different sites from an individual tend to be highly correlated with each other with inter-tumor heterogeneity a substantially more dominant factor. The authors propose that the interactions between tumor cells and microenvironment play an important role in establishing these stable phenotypes and using cell-cell communication analysis stratify the patients for tumor aggressiveness. The communication signatures derived from single-cell analysis is then applied to three large bulk cohorts. The signature scores accurately stratified tumor aggressiveness in each of the three cohorts. Finally, the authors employ RNA-scope to validate some of the predicted cell communication signals and conclude that cell communication plays an important role in establishing a stable tumor phenotype.

My expertise is in single-cell analysis and this review is a critique on the analysis approaches. Overall, this is a very solid manuscript where the authors use appropriate analysis and statistical approaches. The methods section is well written and has sufficient detail for reproducibility. The conclusions and interpretation of the data are done appropriately.

A few additional analysis directions for the authors to consider:

1. How much does the distance between different sections of the same tumor make a difference in assessing intra-tumor heterogeneity?
2. A potential direction of analysis is to explore is to repeat the analysis in Fig. 1B but include all epithelial cells including tumor and normal. The normal cells from different patients should overlap with each other and the presence of matched normals might potentially help further illuminate intra tumor heterogeneity.
3. Similar to point (2), the authors can also consider a patient-by-patient analysis to include tumor and normal epithelial cells from all sites. While it is quite conclusive that the inter-tumor heterogeneity is significantly greater than intra-tumor heterogeneity the extent to which intra heterogeneity might be different across different tumors.

Responses to Reviewers' comments

We would like to express our sincere gratitude to the reviewers for their constructive and helpful comments to improve our paper. We have revised the paper according to the suggestions.

Reviewer #1: (Remarks to the Author): Expert in iCCA and HCC genomics, single-cell genomics, and tumour microenvironment

In this manuscript, Ma, et al. performed a multiregional single-cell transcriptomic analysis of HCC and iCCA from seven liver cancer patients. They validated the results in single-cell data from an additional 37 HCC and iCCA patients, as well as several independent cohorts. An important finding of this paper is that specific molecular network of tumor cells and macrophages (i.e. LGALS9- SLC1A5 and SPP1-PTGER4) represent a stable hub of HCC malignancy. In general, this manuscript would be of interest to the growing crowd of oncologists and those interested in single-cell analysis. However, the majority of the work is descriptive and the lack of validation assays to confirm the annotations dampen the enthusiasm, I raise hereafter would potentially increase its impact.

Major comments:

1. Given the author performed the multi-regional sampling of tumors, it would be interesting to profile the exome profile or computationally infer the copy number to check out the genomic features of tumor subregions.

Response: Thank you so much for the very helpful suggestion. Accordingly, we inferred chromosomal copy number variations (CNVs) from transcriptome and further performed hierarchical clustering of the CNVs for all the malignant cells within each individual patient. We observed relatively homogeneous CNVs across all the tumor cells for some of the cases (e.g., 1H), while relatively heterogeneous tumor cells with distinct CNV clones in others. For example, in the case of 3C, we found 11q gains were enriched in T1 region, although the alterations were also observed in T2 and B regions of the tumor lesion from this case. Noticeably, the differences of CNVs were much greater between patients than within a patient. We did not perform multi-regional single-cell DNA sequencing analysis to measure copy number variations as we think the resolution of somatic copy number variation at the single cell levels is suboptimal. We have revised the manuscript accordingly. Please refer to Supplementary Fig. 4a.

2. Most of the findings of this study are exploratory and lack experimental verification. It is necessary to confirm and validate key findings based on the scRNA-seq discovery e.g. T-cell composition appears stable among different sampling regions or whether LGALS9-SLC1A5 and SPP1-PTGER4 interactions synergistically promote tumor progression. This is very important for the reliability of the findings.

Response: We agree with the reviewer that our current single-cell transcriptomic studies, which is analogous to most of the published single-cell studies in recent days including the latest publication by Garcia-Alonso L et al (Nature 607: 540-47, 2022), are correlative using clinical data and biospecimens. Validation of single-cell transcriptome-based results requires a careful design and comprehensive analysis to establish a causal relationship, as we think experimental

approaches simply using in vitro models to validate the correlative data are suboptimal. These studies should be the subject of future manuscripts. However, we agree with the reviewer that correlative data could be further validated using different methods to strengthen their associations. In our original submission, we validated some of the key findings. For example, the key signaling pairs (LGALS9-SLC1A5 and SPP1-PTGER4) were found to be associated with patient survival using the single-cell data. We then further validated this association as a functional surrogate using RNAscope in situ hybridization on a large cohort (Fig. 7). To further validate some of the data from samples for which single-cell analysis was conducted, we have now performed additional IHC and FACS analyses on the same samples that are still available used in the multiregional single-cell analysis. Specifically, we have performed IHC on T-cell composition and FACS analysis of EPCAM and GPC3 using multiregional samples. These new data are included in Supplementary Fig. 6 and Supplementary Fig. 8. We have revised the manuscript accordingly.

3. In Figure 3, T-cell composition appears stable among different sampling regions, it would be better to further subdivide T cells into different subsets (e.g. Treg, Tex, etc.) and reveal their differences in distribution, functional status, and interactions between tumor border and core regions.

Response: We are very thankful to the reviewer for this great suggestion. We performed clustering of T cells and identified 21 unique subtypes, which were defined based on the top differentially expressed genes in each subset. We further compared the proportions of T-cell subsets among different tumor regions within each case. While T-cell transcriptomic profiles appeared more stable than other cell types (e.g., TAMs), we observed that T-cell states were more dynamic. Among all the cases, 4H had the most stable T-cell subset composition which was consistent with our initial analysis that the highest correlation of T-cell profiles among different tumor regions in 4H was observed (Fig. 3e). In other cases, we found similarity between certain tumor regions while heterogeneity among others. For example, in the case of 1H, we observed relatively stable T-cell state composition in the T1, T2, and T3 regions while distinct states in the B region. This was also revealed from our initial correlation analysis with high correlation scores among the three tumor core regions and low correlation between tumor border and tumor core (Fig. 3e). In addition, we determined the interactions between T-cell subtypes and malignant cells. We found stable communications in some T-cell subtypes, while missing links in others. The results were expected, since not all T-cell subsets occurred in all the tumor regions. Collectively, these results suggest that T cells have more conserved expression profiles than other cell types within each individual case, however, within the T-cell population, cellular states are more dynamic and may vary among tumor regions. Furthermore, we have performed T-cell composition by IHC on multiregional samples and the results were consistent with our scRNA-seq analysis (Supplementary Fig. 8). We have revised the manuscript accordingly. Please refer to Supplementary Figs. 8, 10 and 12.

4. The unsupervised clustering analysis and RNA velocity analysis showed that malignant cells were similar among different tumor regions. Those data were partly inconsistent with previous findings (Ann Oncol. 2019 Jun 1;30(6):990-997.; Gastroenterology. 2022 Jan;162(1):238-252). The authors should provide more evidence to re-confirm their findings and discuss the potential

explanations.

Response: We appreciate the reviewer for this comment. We agree that genomic studies including whole-exome sequencing and single-cell DNA sequencing (scDNA-seq) among liver cancer and other cancer types, have presented evidence about clonal architecture during tumor evolution in individual patient. However, mixed results and interpretation have been seen in recent years. For example, a recent single-cell study (Marjanovic et al, Cancer Cell 2020) of lung tumor evolution using both single-cell RNA sequencing (scRNA-seq) and scDNA-seq found that the clones determined by scDNA-seq are largely independent from the clones with similar cellular states derived from transcriptomic landscape determined by scRNA-seq. These results indicate that genomic alterations may be independent of transcriptomic profiles. This is anticipated because most genomic alterations used for clonality analysis have no functional consequence as they are just passenger events. Consistently, many recent studies have now shown evidence supporting the idea that cellular states defined by scRNA-seq may be better in representing tumor cell clonality and evolution. Obviously, we need to interpret data with caution due to the limitation of the current single-cell technology. We have added a discussion in the revised text accordingly.

5. The number of tumor cells obtained from different regions is too small, which will reduce the accuracy of subsequent studies on the interaction between tumor cells and TME, as well as the issue raised above. For example, 3H and 2H contains only tens of cancer cells. It would be better if the authors utilized FACS or computational strategies to solve this problem.

Response: We appreciate this assessment. Obtaining viable tumor cells from fresh biopsies for single-cell transcriptome analysis is challenging. Due to logistic reasons, we could not isolate many viable tumor cells for single-cell analysis even though we have spent a few years to improve our method. This has been a common theme for most published studies when evaluating human tumor biopsy-based single cells as many could only analyze immune cells which could be survived better during isolation than tumor cells. This is one of the reasons we validated our initial key findings using both our NIH single-cell cohort and further RNAscope analysis with two other larger cohorts.

Nevertheless, we think the comment by the reviewer is valid and we have tried to address this question by both FACS and additional computational strategies. As suggested, we have performed additional experiments using FACS analysis. FACS sorting may alter cellular states, which may not reflect the original states of the cells in patients. This is especially true for biopsy-based strategy. Thus, instead of performing scRNA-seq of tumor cells after FACS sorting, we directly compared the proportion of tumor marker positive cells from multiple tumor locations using FACS sorting. We sorted EPCAM+ cells for one HCC sample and GPC3+ cells for two HCC samples with remaining available cryopreserved single-cell samples from the original 7 patients with multi-region biopsies. We selected these cases because they have elevated EPCAM and GPC3 in tumor cells. These new data are included in Supplementary Fig. 6. We observed from FACS analysis that the proportion of EPCAM+ or GPC3+ cells were similar among the tumor cores while differences are noted in the tumor border region. The ability to detect EPCAM+ or GPC+ cells by a different method further supports our initial study using single-cell transcriptome analysis. However, our initial multi-regional study only included 7 cases and we

will not have statistical power to determine the association between FACS results and those from scRNA-seq.

To further address issues about low tumor cell numbers, we have performed additional computational analyses to test the impact of the number of malignant cells on ligand-receptor pair identification by randomly sampling malignant cells. Specifically, we selected five cases with the highest number of malignant cells in Fig. 5a and randomly sampled 200, 100, 50, 20, and 10 malignant cells from each case for ligand-receptor identification. We performed five times of the random sampling for each setting and further compared the identified pairs with those in Fig. 5a to calculate the accuracy of ligand-receptor pair detection. We found no linear relationship between the number of cells and the accuracy of ligand-receptor interaction determination. However, we did notice a small accuracy drop when the number of malignant cells is 20 (an average accuracy of 80.04%) or 10 (an average accuracy of 71%). We have included these new data in Supplementary Fig. 14.

6. It has been reported that not only tumor cells, but also a subset of macrophages highly express SPP1 (Cell. 2021 Feb 4;184(3):792-809.e23.). Also, two isoforms of SPP1: a secreted form and an intracellular form has been identified with distinct functions (Immunol Res. 2011 Apr;49(1-3):160-72.). So, it is difficult to reveal the true interaction between tumor cells and TAMs through SPP1-PTGER4 simply relying on scRNA-seq data analysis. This should be described more detail to avoid misunderstanding.

Response: We are very thankful to the reviewer for this comment. We agree with the reviewer that SPP1 was mainly expressed in tumor cells and tumor associated macrophages, with a much higher expression in tumor cells in liver cancer (Ye et al, Nat Med 2003). We also recognized that SPP1 contains several isoforms that include both secreted and intracellular isoforms. However, it's difficult to distinguish various SPP1 isoforms with the current 10x genomics single-cell technology. We have revised the discussion in the revised text to clarify these points.

7. What are the overall features of the TME of enrolled patients (i.e. immunosuppressive or immune activated)? Also, the annotation of immune cell subsets is required in Fig. 3a (e.g. Tex, Tem, and etc.). Given the large difference of phenotypic and molecular difference of T cell subsets, it would be better to analyze the specific cell-cell interaction between tumor and T cell subsets in Fig 4a.

Response: Thank you so much for this comment. We generated pseudo-bulk data based on our single-cell profiles and applied the immune signatures in literature (immune features from Sia et al Gastroenterology 2017, cytotoxic and exhaustion features from Zheng et al Cell 2017 and Guo et al Nat Med 2018) to determine the overall features of the tumor microenvironment. In the seven cases enrolled in this multiregional study, we observed immune activation features in 3C and 3H while immune suppressive features in 4H and 1H. For the rest of the patients, lack of the immune activities was observed. In addition, we determined T cell subsets and their interactions with the malignant cells (see response to comment #3). Please refer to Supplementary Figs. 9c, 10 and 12.

8. Inferring the cell-cell interaction only based on CellPhoneDB is not enough. I suggest the

authors to repeat the analysis by using other algorithms (e.g. CellChat) to reconfirm their findings.

Response: Thank you so much for this suggestion. Accordingly, we applied CellChat to study cellular communications to reconfirm our findings. We found > 85% consistency of the ligand-receptor interactions (i.e., SPP1-PTGER4 and LGALS9-SLC1A5) between tumor cells and tumor associated macrophages using the two methods. Please refer to Supplementary Fig. 14a.

Minor comments:

1. The process of sample acquisition should be described in more detail, such as the size of tumor pieces, the distance between tumor border and core, and whether the three regions of tumor core are adjacent.

Response: Thank you for this comment and we have revised the texts to describe the details. A total of seven primary liver cancer patients treated at the University Medical Center in Mainz and the NIH Clinical Center in Bethesda, have been enrolled prospectively into this study. Among them, three patients were diagnosed with iCCA and four were diagnosed with HCC. Tumor size for each patient can be found in Table S1. All patients received surgical resection. A total of five samples from the tumor core, tumor border and adjacent non-tumor tissue were collected for each patient. Specifically, we collected three samples from the tumor core that were not adjacent, one sample at the tumor border, and one sample from the adjacent non-tumor tissue that was not locally close to the tumor. Each sample was measured about 5 mm diameter in size before single-cell library preparation. Sample collection was performed with informed consent from patients. We have revised the manuscript to clarify these points. Please refer to the Human sample collection part of the Methods section.

2. The authors should specify the count of UMIs/genes of detected cancer cells.

Response: Thank you for the comment. An average of 3,106 genes and 16,658 UMIs were detected per malignant cell. Please refer to the Separation of malignant cells and non-malignant cells part of the Methods section in the revised text.

3. Not all the tumor samples and regions were enrolled for the scRNA-seq analysis (2C, 1CT2, etc.), Please explain this in the main text.

Response: We apology for our initial failure in describing the usage of samples in analyzing malignant cells. Specifically, we collected 5 samples (i.e., three tumor cores, one tumor border and an adjacent normal tissue) from each of the 7 enrolled patients. For the patient 3C, we removed one sample due to single-cell library failure and thus a total of 34 samples were included in this study. All the samples were used for studying non-malignant cells. In the analysis of malignant cells, we only used the samples with >10 malignant cells detected. With this criterion, we didn't detect enough malignant cells in 2C as well as the samples labeled as N.D. (not detected) in Figure 1f. We have revised the manuscript to describe the usage of samples more clearly. Please refer to the Multiregional liver tumor cell transcriptome profiles part of the Results section and the Separation of malignant cells and non-malignant cells part of the Methods section.

4. The authors should provide more technical description on the data integration of single cells from different patients.

Response: Thank you so much for the comment. We used the cellranger count pipeline to process fastq files and further applied cellranger aggr pipeline to perform read depth normalization of all the samples. This method is performed based on the average reads per cell mapped confidently to the targeted transcriptome to avoid artifacts that may be introduced due to differences in sequencing depth among samples. It's a standard way of normalization for single-cell profiles from the samples sequenced in one batch, which is perfect for our data. We didn't incorporate other methods, e.g., Harmony (more suitable for different datasets or from different platforms), which may introduce artifacts for our data due to over normalization. We have described the data integration process in a more detailed way in the revised text. Please refer to the scRNA-seq data pre-processing part of the Methods section.

5. The authors should discuss the potential clinical impact of their paper.

Response: We greatly appreciate the reviewer for the comment. In the single-cell studies of different cancer types, the phenomenon of extensive tumor heterogeneity has been noticed, which creates a major barrier for effective cancer interventions. Sampling bias could be an issue when one uses a single biopsy to determine tumor biology and response to treatment. Thus, in clinical practice, it is important to identify features that are relatively stable and can be used to assess molecular features of a tumor during the course of clinical intervention to avoid sampling bias. In this study, we found the communications between tumor cell and tumor associated macrophages represent a stable hub for HCC malignancy, which may open a path for therapeutic exploration. This is consistent with the notion that tumor cells continuously communicate with the tumor microenvironment, defining the molecular map underlining tumor biodiversity may be a key to improve our understanding of tumor heterogeneity and further identifying novel therapeutic targets. Our study provides both a conceptual advance and potential clinical impact on liver cancer studies. We have added a discussion about the clinical impact in the revised manuscript. Please refer to the Discussion section.

Reviewer #2 (Remarks to the Author): Expert in hepatocellular carcinoma tumour immune microenvironment and immunology

The manuscript by Ma et al. performed multiregional single-cell transcriptome profiling of 7 liver cancer patients (4 HCC and 3 iCCA patients), validated using single-cell seq data in additional 37 HCC and iCCA patients and identified specific molecular network/fingerprint consisting of LGALS9-SLC1A5, SPP1-PTGER4 as tumor and macrophage-derived ligand-receptor interaction pairs, linked to tumor aggressiveness. Independent validation of these interaction pairs was performed using bulk transcriptome profiles of 542 HCC samples from 3 independent cohorts and multiplexed fluorescence in situ hybridization-based profiles of 258 HCC samples from 2 cohorts. The study consists of deep and comprehensive immunoprofiling at single-cell levels and large validation datasets, showing important findings on a stable molecular network that could be targeted therapeutically. However, in order to support the above claims,

there are multiple areas where data presented will require enhanced description and explanation. Also, some further validation data could be provided.

Major comments:

1. Spatial single-cell analysis is typically linked to spatial transcriptomic analysis with matched histochemistry data and since the authors have in fact performed “multiregional single-cell transcriptomic profiling” on 7 liver cancer patients (4 HCC and 3 iCCA patients), while majority of the validation were performed on single-cell data (37 liver cancer patients) and other databases, I would recommend to rephrase the title of the manuscript e.g. to remove the word “Spatial”, to more appropriately reflect the main findings of the current study.

Response: Thank you so much for the comment. Since we collected tumor specimens from multiple locations within a tumor lesion, we used “spatial” to emphasize on the stable features (lock-and-key) among different spatial locations. Those features may reflect the intrinsic tumor biology and represent tumor aggressiveness. We agree that we may use interpretation as the conclusion in the title. Accordingly, to avoid misunderstanding of spatial transcriptomic analysis, e.g., 10x Visium, we have changed “spatial” to “multiregional” in the title of the manuscript.

2. Fig. 1, why is the data from sample 2C missing? In that case, should the number of samples be revised to only 6 liver cancer patients (4HCC + 2 iCCA)? Also it seems like not all patients consistently have data from 3 tumour cores T1-T3 and 1 tumor border (B), this should also be indicated clearly in the results. Next, are each dataset normalized by its respective Normal tissues as control? It will be good to show all the data with Normal vs Tumours vs Border as Normal would be expected to be distinctly different from tumours. It is not known how different Tumour Cores vs Border should be distinctive, hence it will be good to check if N is actually separated from T.

Response: We appreciate the reviewer for this comment and apologize for our initial failure in describing the samples clearly. Specifically, we collected 5 samples (i.e., three tumor cores, one tumor border and an adjacent normal tissue) from each of the 7 enrolled patients (4 HCC and 3 iCCA). For the patient 3C, we removed one sample due to single-cell library failure and thus a total of 34 samples were included in this study. All the samples were used for studying non-malignant cells. In the analysis of malignant cells, we only used the samples with >10 malignant cells detected. With this criterion, we didn’t detect enough malignant cells in 2C as well as the samples labeled as N.D. (not detected) in Figure 1f. We have revised the manuscript to describe the samples more clearly. In addition, we performed tSNE analysis of epithelial cells from both tumor and normal regions. As expected, malignant cells formed patient-specific clusters while normal epithelial cells from different patients were mixed. We observed two major clusters of hepatocytes and cholangiocytes respectively, suggesting distinct functional states within each cell type which is consistent with published studies in normal liver tissue (MacParland et al, Nat Commun 2018). We have revised the manuscript accordingly. Please refer to Supplementary Fig. 2, the Multiregional liver tumor cell transcriptome profiles part of the Results section, and the Separation of malignant cells and non-malignant cells part of the Methods section.

3. It is not clear how heterogeneity scoring is determined, based on RNA expression of each single cell? All genes or selected genes? This information is not immediately clear from the

description which makes interpretation of comparison between inter- or intra-tumoural heterogeneity difficult.

Response: We are very thankful to the reviewer for this comment to help us improve our method description. Heterogeneity was determined based on the pair-wise correlation of malignant cells within a tumor region (intraregional heterogeneity), across different regions within a case (interregional heterogeneity), and across multiple patients (intertumor heterogeneity). We applied the top 2,000 most variable genes of all the malignant cells from all patients in order to have a fair comparison. The distribution of the correlation values was used to indicate tumor heterogeneity (Fig. 1f). In other words, the correlation was determined at single-cell level with the same set of 2,000 genes applied to all the correlation calculations. Please refer to the Tumor Heterogeneity part of the Methods section for a detailed description.

4. Are there any distinctive differences between HCC vs iCCA?

Response: Thank you for this comment. HCC and iCCA are two histological subtypes of liver cancer. Bulk transcriptomic profiling of ~200 HCC and iCCA patients (Chaisaingmongkol et al Cancer Cell 2017) indicate that the two clinical subtypes have both distinct and shared molecular features. In our multiregional patient cohort, the malignant cells from HCC (4 patients) and iCCA (2 patients) were separated based on hierarchical clustering method (Fig. 1d), indicating distinct transcriptomic profiles among the two clinical subtypes in this cohort. However, when evaluating the communications between malignant cells and the tumor microenvironment, mixed features were observed (Fig. 4d). Moreover, in the analysis of an addition single-cell cohort, we found Cluster 1 comprised both HCC and iCCA while Cluster 2 was mainly composed of HCC (Fig. 5a). Collectively, these results suggest that HCC and iCCA have both distinct and common molecular features, which is consistent with published studies. We have added a discussion about this point in the revised text. Please refer to the Discussion section.

5. Fig. 2 showed only one sample 1C while Fig. S3 showed another 3 (3C, 1H and 4H), in fact data from all samples should be shown. The trend are in fact not very obvious for other cases and it is not clear why not the same genes were shown in Fig. 2 vs Fig. S3.

Response: Many thanks again to the reviewer for pointing out our initial failure in describing the samples clearly. To study the dynamics of malignant cells, we used the RNA velocity method from the scvelo Python package. However, this method failed to recover the full dynamics of malignant cells within a case if the number of cells is too small. Thus, in our initial analysis, we only determined the velocity for the four cases (i.e., 1C, 3C, 1H, and 4H) with > 50 malignant cells detected in each individual case. We have revised the method part to state this point clearly. The purpose of RNA velocity analysis is to demonstrate the similarities among different tumor regions within each individual case. Along the determined latent time, cells from multiple tumor regions were mixed, indicating similarity among tumor regions. Since the expression of stemness related makers may vary among malignant cells within a case, we don't expect a perfect downtrend along the latent time for all the cases. In other words, certain stemness related genes may be expressed in a group of malignant cells while other stemness related genes may be expressed in a distinct group of malignant cells within a case. For the selection of tumor stemness related genes and tumor evolution related genes, we applied 10 genes including

EPCAM, KRT19, ICAM1, PROM1, LGR5, CD44, ANPEP, HNF4A, ALDH1A1, SPP1 in our analysis. During RNA velocity analysis of malignant cells in each case, the low expressed genes were removed using the function `scv.pp.filter_and_normalize(data, min_shared_counts=20, n_top_genes=2000)` in order to select abundant genes to recover the cellular dynamics. Thus, not all the 10 genes were found in latent time analysis of each individual case. We only included those remaining genes in each case in Fig. 2b and Supplementary Fig. 5. This is also consistent with literature of heterogeneity in tumor stemness genes among tumor cells (Zheng et al Hepatology 2018). We have added more description of this part in the revised text. Please refer to the Cellular dynamics determined by RNA velocity method part of the Methods section.

6. Could the lower intratumoural heterogeneity of T cells due to more equal physical distribution within the TME where the other cell types may show more heterogenous distribution in TME hence leading to higher heterogeneity? In another word, is the heterogeneity physical or genetic/phenotypic? As the stable molecular feature is the key focus of the study, it is essential to determine this e.g. By IHC analysis of each of these subsets in the TME.

Response: Many thanks for this comment. We agree that these are interesting and important questions that deserve careful experimentations. However, available data in the current study and the literature are insufficient to address these questions thereby conclusive statements. We have discussed this point in our revised manuscript.

7. If T cells are the most stable subset within the TME, why is the subsequent stable ligand/receptor pairs were identified between malignant/TAM instead? How stable and heterogeneity of TAM in TME?

Response: Thanks for raising this question. We found T cells profiles were more stable than other cell types among different tumor regions. We also detected interactions between tumor cells and T cells in our tumor-TME communication analysis (Figs. 4a and 5a). In our clustering analysis of ligand-receptor interactions, we did find that several tumor-T cell interactions were elevated in Cluster 1 which had worse patient outcome (Figs. 5a and 5d), indicating that these interactions were also related to tumor aggressiveness. The association between the significant ligand-receptor interactions pairs and tumor prognosis including T-cell based pairs was validated in three independent HCC cohorts of bulk transcriptome data. We selected the top two interaction pairs between tumor and TAMs (i.e., SPP1-PTGER4, LGALS9-SLC1A5) in Cluster 1 for RNAscope validation for proof of principle analysis, as validation of all significant pairs using RNAscope would be technical challenging and require substantial resources. It is noted that TAMs were also relatively homogeneous in case 3H, 4H while more heterogeneous in the rest of the cases in our multiregional patient cohort. Nevertheless, we validated our single-cell data using RNAscope about the communications between tumor and TAMs (SPP1-PTGER4, LGALS9-SLC1A5). Further studies to establish a causal relationship among tumors and immune/stromal cells will require a careful planning using appropriate experimental models. We have discussed these points to make it clear.

8. Fig. 4C, how is regional stability calculated? Data shown in Fig.4E & 4F is interesting, though most likely due to a more stable interpatient TME compared to malignant cells (Fig. 1 vs Fig. 3).

Response: Thanks so much to the reviewer for this comment. Regional stability was calculated based on the ligand-receptor pairs occurred in each individual case. Among all the identified ligand-receptor pairs, we determined the proportion of the pairs that were found in all the tumor regions or part of the tumor regions. For example, in the case of 1H, 9 pairs were identified in total (Fig. 4a). Among those pairs, 7 (77.8%) were found in all the tumor regions and 2 (22.2%) were found in part of the tumor regions in this case. We have added a more detailed description in the revised text. Please refer to the Communication of malignant cells and the TMEs part of the Methods section. In addition, we totally agree with the reviewer on the thoughts about switching tumor and TME. These results suggest that tumor is the driver in a tumor ecosystem.

9. SPP1 is well known to be macrophages-specific genes however according to their CellPhoneDB data, it is expressed on tumour binding to PTFER4 expressed on TAMs. In my opinion, it is important to determine where exactly is SPP1 expressed in HCC tumours e.g. by IHC.

Response: We agree that earlier studies demonstrated that SPP1 is expressed in macrophages. However, many subsequent studies including our earlier publication (Ye QH et al, Nat Med 9: 416-23, 2003) among others demonstrated that SPP1 is also highly elevated in many tumor types including HCC. Since the specific clone of anti-SPP1 antibody used in our earlier study is no longer available, we tested two additional anti-SPP1 antibodies (i.e., Company: MyBioSource, Cat.no. MBS4382252, Product name: Osteopontin (OSP)/Secreted Phosphoprotein 1 (SPP1) Monoclonal Antibody [Clone OSP/4589] (Rat); Company: R&D systems, Cat.no. AF1433, Product name: Human Osteopontin/OPN Antibody (Goat)) using paraffin tissues from the original tissues used for multi-regional scRNA-seq analysis. OPN antibody AF1433 did not work at all. We were able to detect SPP1 expression in tumor cells using IHC with the expression patterns consistent with previously published data (Ye QH et al, Nat Med 2003). We have included these data as Supplementary Fig. 7.

10. Validation shown in Fig. 7 should be done using IHC instead to show the actual protein expression and ligand/receptor interaction. Also, no tumour or TAM markers were used together to specify the sources of these markers.

Response: We agree with the reviewer on this comment and have made an effort to perform additional IHC experiments. We selected PTGER4 and SPP1 for IHC and tested several different commercially available antibodies. We were able to detect SPP1 expression in tumor cells using one commercial antibody (see our response to comment #9 above). However, we failed to detect PTGER4. We have attempted two antibodies against PTGER4, a protein that has not been studied much in literature. These include anti-PTGER4 (rabbit; Sigma; Cat# HPA011226) and anti-PTGER4 (mouse; Proteintech; Cat# 66921-IG). We have exhaustively evaluated various IHC conditions independently both at the laboratories in the NCI and in Mainz, Germany. Both laboratories failed to detect specific signals in all available FFPE tissues used for the original multi-regional scRNA-seq analysis. Therefore, we will not be able to evaluate PTGER4 at the protein levels.

We hope the reviewer would also appreciate the complexity of IHC for precision quantification, as we think IHC is often technically challenging, and the results are largely dependent on

availability of high-quality antibodies. This is even more challenging when targeting different proteins with multiple antibodies were performed on the same slide using multi-channel chromogenic detection. For this reason, our initial effort was mainly relying on the use of RNAscope analysis as a proof of principle experiment even though this approach has its limitation and therefore need to reach interpretation with caution. We have discussed these points in the Discussion section of the revised manuscript as a limitation of this study.

Minor comment:

1. I have noticed some discrepancies in number of cases or samples involved in the data presented, please check to verify the accuracy of reported numbers or to explain why if any data have been omitted.

Response: Thank you so much again for this comment. We have revised the manuscript to describe the samples more clearly. Please also see our responses to comments #2 and #5.

Reviewer #3 (Remarks to the Author): Expert in single-cell RNA-seq analysis

In this manuscript, the authors aim to characterize the characteristics and intra-tumor and inter-tumor heterogeneity using single-cell RNA-seq profiling of multiple spatial sites of the same tumor across multiple tumors in hepatocellular carcinoma. Analysis of tumor cells demonstrates that the tumor cells from the different sites from an individual tend to be highly correlated with each other with inter-tumor heterogeneity a substantially more dominant factor. The authors propose that the interactions between tumor cells and microenvironment play an important role in establishing these stable phenotypes and using cell-cell communication analysis stratify the patients for tumor aggressiveness. The communication signatures derived from single-cell analysis is then applied to three large bulk cohorts. The signature scores accurately stratified tumor aggressiveness in each of the three cohorts. Finally, the authors employ RNA-scope to validate some of the predicted cell communication signals and conclude that cell communication plays an important role in establishing a stable tumor phenotype.

My expertise is in single-cell analysis and this review is a critique on the analysis approaches. Overall, this is a very solid manuscript where the authors use appropriate analysis and statistical approaches. The methods section is well written and has sufficient detail for reproducibility. The conclusions and interpretation of the data are done appropriately.

A few additional analysis directions for the authors to consider:

1. How much does the distance between different sections of the same tumor make a difference in assessing intra-tumor heterogeneity?

Response: We are very thankful to the reviewer for this comment. The tumor size varies among 7 patients in this multiregional single-cell cohort with the largest tumor size of 24.5 cm in diameter for 3H and the smallest size of 3 cm in diameter for 4H. We performed pair-wise correlation analysis of all available malignant cells from different tumor regions within a case

and used the mean value to reflect tumor heterogeneity. We further calculated its correlation with tumor size. We observed no significant correlation in our cohort. In addition, we calculated the ratio of tumor heterogeneity between tumor border and tumor core as well as the heterogeneity across tumor regions. We didn't find a relationship with tumor size for both the measurements as well. We have revised the manuscript to clarify this point. Please refer to Supplementary Figs. 4b-4d.

2. A potential direction of analysis is to explore is to repeat the analysis in Fig. 1B but include all epithelial cells including tumor and normal. The normal cells from different patients should overlap with each other and the presence of matched normal might potentially help further illuminate intra tumor heterogeneity.

Response: We appreciate the reviewer for the very helpful comment. Accordingly, we performed tSNE analysis of epithelial cells from both tumor and adjacent normal tissues. As expected, malignant cells formed patient-specific clusters while normal epithelial cells from different patients were grouped together. We observed two major clusters of hepatocytes and cholangiocytes respectively, suggesting distinct functional states within each cell type which is consistent with published studies in normal liver (MacParland et al, Nat Commun 2018). Please refer to Supplementary Fig. 2.

3. Similar to point (2), the authors can also consider a patient-by-patient analysis to include tumor and normal epithelial cells from all sites. While it is quite conclusive that the inter-tumor heterogeneity is significantly greater than intra-tumor heterogeneity the extent to which intra heterogeneity might be different across different tumors.

Response: Thank you so much for this comment. We performed analysis of epithelial cells for each patient separately. Consistent with the results from all cases, epithelial cells from normal tissues were largely separated from malignant cells in each individual case. Compared with malignant cells, hepatocytes or cholangiocytes were generally more homogeneous in most of the cases. Please note that we applied tSNE for most of the cases while hierarchical clustering for two cases due to failure in generating tSNE with a small number of cells in the two cases. The results can be found in Supplementary Fig. 2.

REVIEWERS' COMMENTS

Reviewer #1 (Remarks to the Author):

In this revised version, Ma, et al. performed additional FACS/mIHC analysis and solved most of my concerns. I have no further comments.

Reviewer #2 (Remarks to the Author):

The authors have made substantial improvements on the clarity of the manuscript and efforts to provide additional supporting data. I have no further comments.

Reviewer #3 (Remarks to the Author):

The authors have satisfactorily addressed the comments I raised in my initial review and I support the publication of this manuscript in Nature Communications.

Responses to Reviewers' comments

Reviewer #1 (Remarks to the Author):

In this revised version, Ma, et al. performed additional FACS/mIHC analysis and solved most of my concerns. I have no further comments.

Response: We are very thankful to the reviewer for all the constructive comments to help us improve our manuscript.

Reviewer #2 (Remarks to the Author):

The authors have made substantial improvements on the clarity of the manuscript and efforts to provide additional supporting data. I have no further comments.

Response: Many thanks to the reviewer.

Reviewer #3 (Remarks to the Author):

The authors have satisfactorily addressed the comments I raised in my initial review and I support the publication of this manuscript in Nature Communications.

Response: Thank you so much. We do appreciate that.